# HEIST: A GRAPH FOUNDATION MODEL FOR SPATIAL TRANSCRIPTOMICS AND PROTEOMICS DATA

**Hiren Madhu, João Felipe Rocha, Tinglin Huang, Siddharth Viswanath,**
**Smita Krishnaswamy**\*, **Rex Ying**\*
Yale Univeristy, USA    \*Co-Senior Author

Corresponding Authors: {smita.krishnaswamy, rex.ying}@yale.edu
🤗 HirenMadhu/HEIST

## ABSTRACT

Single-cell transcriptomics and proteomics have become a great source for data-driven insights into biology, enabling the use of advanced deep learning methods to understand cellular heterogeneity and gene expression at the single-cell level. With the advent of spatial-omics data, we have the promise of characterizing cells within their tissue context as it provides both spatial coordinates and intra-cellular transcriptional or protein counts. Beyond transcriptomics, proteomics offers a complementary view by directly measuring proteins, which are the primary effectors of cellular function and key therapeutic targets. However, existing models either ignore the spatial information or the complex genetic and proteomic programs within cells. Thus they cannot infer how cell internal regulation adapts to microenvironmental cues. Furthermore, these models often utilize fixed gene vocabularies, hindering their generalizability to datasets with different genes than pretraining. In this paper, we introduce HEIST, a hierarchical graph transformer foundation model for spatial transcriptomics and proteomics. HEIST models tissues as hierarchical graphs. The higher level graph is a spatial cell graph, and each cell in turn, is represented by its lower level gene co-expression network graph. Rather than using a fixed gene vocabulary, HEIST computes gene embeddings from its co-expression network and cellular context. HEIST achieves this by performing both intra-level and cross-level message passing to utilize the hierarchy in its embeddings and can thus generalize to novel datatypes including spatial proteomics without re-training. HEIST is pretrained on 22.3M cells from 124 tissues across 15 organs using spatially-aware contrastive and masked autoencoding objectives. Unsupervised analysis of HEIST embeddings reveals spatially informed subpopulations missed by prior models. Downstream evaluations demonstrate generalizability to proteomics data and state-of-the-art performance in clinical outcome prediction, cell type annotation, and gene imputation across multiple technologies.

## 1 INTRODUCTION

Single-cell RNA sequencing (scRNA-seq) has revolutionized our ability to study gene expression at the resolution of individual cells, offering data-driven insights into biology. The complexity of these datasets has fueled the development of machine learning methods for modeling cellular diversity, predicting cell states, and imputing or denoising gene expression values (Cui et al., 2024; Hu et al., 2021; Grønbech et al., 2020; Wen et al., 2023a; Bravo Gonzalez-Blas et al., 2024; Dijk et al., 2017). However, a limitation of scRNA-seq is the lack of spatial context of cells within tissues, which is important for understanding processes such as tissue organization, microenvironment interactions and how gene co-expression influences tissue-level behaviors. Spatial transcriptomics is an emerging technology that bridges this gap by preserving the physical locations of gene expression measurements, enabling a holistic study of tissue architecture, cell-cell communication, and their aberrations in the tumor context (Xiao & Yu, 2021; Rodriques et al., 2019). Similarly, spatial proteomics assays were developed to directly capture protein abundance, expression and signaling pathways, offering a complementary layer of biological insight. Despite advances in spatial omics, datasets remain limited by low throughput, platform and tissue heterogeneity, and scarce labels, often necessitating

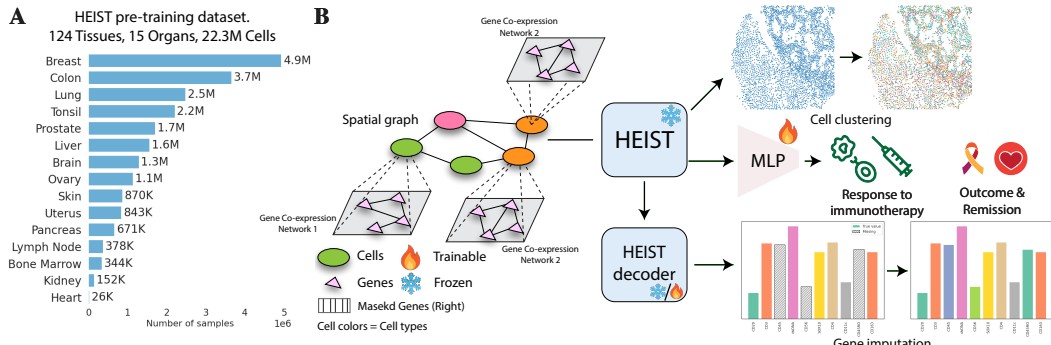

Figure 1: **Overview of the HEIST framework.** (A) HEIST is pre-trained on a large-scale spatial transcriptomics dataset spanning 124 tissues and 15 organs (22.3M cells). (B) HEIST encodes both gene co-expression networks and spatial cell graphs to support downstream tasks such as cell clustering, gene imputation, and clinical outcome prediction (e.g., immunotherapy response, remission). The HEIST decoder can be fine-tuned while the encoder remains frozen.

dataset-specific models (e.g., MIBI (Angelo et al., 2014), Imaging CyTOF). A foundation model trained on diverse spatial omics can address these challenges by learning generalized representations across tissues, organs, settings, and technologies, enabling strong performance on downstream tasks even with limited data.

To this end, we propose HEIST (**H**ierarchical **E**mbedd**I**ngs for **S**patial **T**ranscriptomics), the first foundation model for spatial transcriptomics that explicitly models both spatial proximity and internal co-expression networks while also enabling cross-modal transfer to proteomics. Through cross-level message passing, internal embeddings are shaped by the spatial context of their parent cell, while cell embeddings are updated from their constituent genes. This produces adaptive representations, allowing the same gene to be encoded differently depending on context. In this way, HEIST unifies spatial and molecular hierarchies, linking gene-level interactions to tissue-level phenotypes.

Prior foundation models such as SCGPT (Cui et al., 2024), SCFOUNDATION (Hao et al., 2023), and CELLPLM (Wen et al., 2023a) either neglect cell–cell structure or are limited to predefined gene sets, hindering generalization to unseen genes and proteins. Graph-based methods like GRAPHST (Long et al., 2023) and STAGATE (Dong & Zhang, 2022) (Explained in detail in Appendix A) capture spatial neighborhoods but remain task-specific and non-transferable. While transformer-based models like SCGPT-SPATIAL (Wang et al., 2025) treat all genes as fully connected graph and fail to model the inductive bias of co-expression. HEIST overcomes these gaps by associating each cell with a co-expression network that interacts with its spatial neighbors, incorporating microenvironmental cues into representations.

We pretrain HEIST (Figure 1(A)) on 22.3 million cells from 124 tissues across 15 organs and two technologies, and evaluate on four downstream tasks (Figure 1(B)): clinical outcome prediction, cell type annotation, gene imputation, and cell clustering. We achieve state-of-the-art performance across seven organs. HEIST enables the discovery of spatially-informed cellular subpopulations that previous models fail to do and is **8× faster** than SCGPT-SPATIAL and **48× faster** than SCFOUNDATION. We summarize our contributions as follows:

- **Modeling inter-cellular and hierarchical effects of co-expression networks:** HEIST is the first foundation model for spatial omics to explicitly incorporate co-expression networks alongside spatial graphs in a hierarchical graph, enabling a local gene programs to influence tissue-level organization, and vice versa.

- **Hierarchical representation learning with biological inductive bias**: Using biologically motivated hierarchical modeling, HEIST captures fine-grained gene co-expression within cells and long-range cellular interactions through novel cross-level message passing, producing biologically contextualized embeddings.

- **A task-agnostic, general-purpose foundation model**: HEIST is trained in a self-supervised manner on a large-scale corpus of spatial transcriptomics data comprising over 22.3M cells spanning 15 organs and 124 tissues. In downstream evaluations, HEIST achieves state-of-the-art

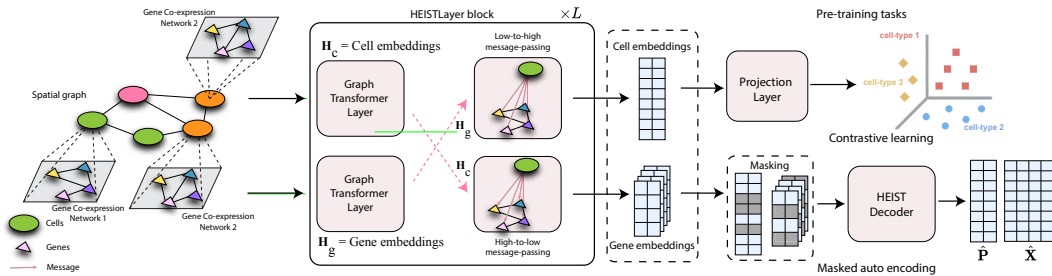

Figure 2: **HEIST Architecture. HEIST Layer block presented in Figure A2.**

performance across four diverse tasks—outperforming prior models, generalizes to proteomics data, while being computationally efficient.

## 2 METHOD

In this section, we describe the architecture of HEIST, a hierarchical graph transformer designed to learn multi-level embeddings for spatial transcriptomics and proteomics data. HEIST takes as input, a set of graphs $\{\mathcal{G}_c(\mathcal{C}, \mathcal{E}, \mathbf{P}, \mathcal{T}), \{\mathcal{G}_g^{t_k}(\mathcal{V}, \mathcal{E}_{t_k}, \mathbf{X}_k)\}_{k=0}^{|\mathcal{C}|}\}$, where $\mathcal{G}_c$ is a spatial graph capturing spatial proximity, $\mathcal{C}$ is the set of cells, $\mathcal{E}$ is the set of spatial edges, $\mathbf{P} \in \mathbb{R}^{|\mathcal{C}| \times 2}$ represents the spatial positions, $\mathcal{T}$ is the set of cell types, and $t_k$ is cell-type of cell $k$. Each graph $\mathcal{G}_g^{t_k}$ represents a gene co-expression network within cell $k$, $\mathcal{V}$ is the set of genes, and $\mathcal{E}_{t_k}$ and $\mathbf{X}_k$ denote the edges of the gene co-expression network and expression values for cell $k$, respectively.

**Hierarchical Graph Construction.** As shown in Figure A1, we first preprocess the data by removing outliers, normalizing gene expression, and retaining highly variable genes. Then we apply MAGIC (Dijk et al., 2017) to denoise gene expression values and reduce dropout noise. To build the gene co-expression networks, we first subset the cells based on cell-types using provided annotations or Leiden clustering. Following this, we compute pairwise mutual information between denoised genes within each type, and connect gene pairs above a threshold $\tau$. This results in total of $|\mathcal{T}|$ gene co-expression networks with mutual information prior. We create the spatial cell–cell graph by computing Voronoi polygons from cell coordinates and connecting cells in adjacent polygons. We then connect each cell with the gene co-expression network of that cell-type. The resulting outputs are a spatial graph $\mathcal{G}_c(\mathcal{C}, \mathcal{E}, \mathbf{P}, \mathcal{T})$ and a set of gene co-expression networks $\mathcal{G}_g^{t_k}(\mathcal{V}, \mathcal{E}_{t_k}, \mathbf{X}_k)_{k=1}^{|\mathcal{C}|}$. We discuss the graph creation at length in Appendix B. Given these graphs, HEIST computes cell embeddings and gene embeddings $\mathbf{Z}_c \in \mathbb{R}^{|\mathcal{C}| \times d}$ and $\mathbf{Z}_g \in \mathbb{R}^{|\mathcal{C}||\mathcal{V}| \times d}$, such that

$$\mathbf{Z}_c, \mathbf{Z}_g = \text{HEIST}\left(\mathcal{G}_c(\mathcal{C}, \mathcal{E}, \mathbf{P}, \mathcal{T}), \{\mathcal{G}_g^{t_k}(\mathcal{V}, \mathcal{E}_{t_k}, \mathbf{X}_k)\}_{k=0}^{|\mathcal{C}|}\right).$$

As shown in Figure 2, the model first performs intra-level message passing (Equation 1) within each graph, followed by cross-level message passing (Equation 2) to integrate multi-modal information. HEIST is pre-trained using a combination of contrastive and auto-encoding objectives on gene expression and cell locations. By using these components, HEIST can learn expressive and context-aware cell and gene embeddings that reflect biologically meaningful relationships between cells and genes. Note that as a result of this setup, gene representations are themselves learned in the context of the hierarchical graph, instead of based on a fixed gene vocabulary. They are initialized with rank-based and sinusoidal positional encodings, and dynamically updated through message passing in co-expression graphs, allowing HEIST to generalize to unseen genes or proteomic features by grounding embeddings in co-expression dynamics.

### 2.1 HEIST ARCHITECTURE.

First we initialize the input cell embeddings $\mathbf{H}_c^{(0)}$ and gene embeddings for cell $k$ $\mathbf{H}_g^{k^{(0)}}$ using positional encodings explained in Appendix C.1. Then they are passed through `HEISTLayer` $L$ times, and the representations are calculated using the equation below:

$$\mathbf{H}_c^{(l)}, \mathbf{H}_g^{(l)} = \text{HEISTLayer}(\mathbf{H}_c^{(l-1)}, \mathbf{H}_g^{(l-1)}, \mathcal{E}, \{\mathcal{E}_{t_k}\}_{k=0}^{|\mathcal{C}|})$$

`HEISTLayer` is divided into two steps, intra-level message passing and cross-level message passing. We perform intra-message passing and calculate the intermediate representations as shown in equation below:

$$\tilde{\mathbf{H}}_c^{(l)} = \texttt{CellGraphTransformer}(\mathbf{H}_c^{(l-1)}, \mathcal{E}), \tilde{\mathbf{H}}_g^{(l)} = \texttt{GeneGraphTransformer}(\mathbf{H}_g^{(l-1)}, \{\mathcal{E}_{t_k}\}_{k=0}^{|\mathcal{C}|})$$
(1)

where `CellGraphTransformer` and `GeneGraphTransformer` are graph transformers as explained in the Appendix Section C.2.

**Cross-level message passing.** To integrate spatial and gene modalities, HEIST performs cross-level message passing between cell and gene graphs at each layer (Figure 2). Gene embeddings are updated based on spatial context through their parent cell's embedding, while cell embeddings are refined using pooled summaries of their genes, ensuring transcriptional states shape spatial identity. This bidirectional interaction captures tissue hierarchy, where gene expression depends on both co-expression signals and spatial microenvironments. HEIST thus learns representations that reflect both local gene co-expression and large-scale tissue structures.

We use the directional attention mechanism to perform cross message passing as shown in equation below:

$$\texttt{CrossMessagePassingLayer}(\mathbf{H}_{\text{to}}, \mathbf{H}_{\text{from}}) = \left( \frac{< \mathbf{H}_{\text{to}}\mathbf{W}_q, \mathbf{H}_{\text{from}}\mathbf{W}_k >}{\sqrt{d}} \right) \cdot (\mathbf{H}_{\text{to}}\mathbf{W}_v), \quad (2)$$

where $\mathbf{W}_q, \mathbf{W}_k, \mathbf{W}_v \in \mathbb{R}^{d \times d}$ are learned weight matrices, and $< \cdot, \cdot >$ is the row-wise inner product. Let $\tilde{\mathbf{H}}_g^{(l)}$ and $\tilde{\mathbf{H}}_c^{(l)}$ denote the intermediate gene and cell embeddings at layer $l$ after intra-level updates and before cross-level integration. HEIST updates these representations using the equation below:

$$\mathbf{H}_g^{(l)} = \texttt{CrossMessagePassingLayer}(\tilde{\mathbf{H}}_g^{(l)}, \tilde{\mathbf{H}}_c^{(l)\text{repeat}}), \quad \mathbf{H}_c^{(l)} = \texttt{CrossMessagePassingLayer}(\tilde{\mathbf{H}}_c^{(l)}, \bar{\mathbf{H}}_g^{(l)}),$$

where $\tilde{\mathbf{H}}_c^{(l)\text{repeat}} \in \mathbb{R}^{|\mathcal{C}||\mathcal{V}| \times d}$ represents the cell embeddings repeated $|\mathcal{V}|$ times to align with the gene embeddings in each cell. Each gene receives information from its parent cell, enabling spatial context to modulate gene-level representations. Conversely, $\bar{\mathbf{H}}_g^{(l)} \in \mathbb{R}^{|\mathcal{C}| \times d}$ is obtained by aggregating the gene embeddings within each cell: $\bar{\mathbf{H}}_g^{(l)} = [\texttt{AGG}(\tilde{\mathbf{H}}_g^1), \dots, \texttt{AGG}(\tilde{\mathbf{H}}_g^{|\mathcal{C}|})]$, where $\texttt{AGG}(\cdot)$ can be aggregation function such as `MEAN` pooling or differential pooling (Ying et al., 2018). This allows the cell embedding to be informed by the internal transcriptional state of the cell.

**Advantages.** This formulation enables targeted, direction-aware communication between modalities while preserving their structure and semantics. It is well suited for spatial transcriptomics and proteomics data, where cell and gene representations must be coupled yet retain distinct meanings. By letting each gene attend to its parent cell embedding, HEIST shapes gene representations in a cell-specific, spatially informed way, and vice versa. Unlike symmetric attention or feature concatenation, this directional mechanism preserves data hierarchy, respects modality roles, and captures how local gene co-expression drives tissue organization. Directional message passing respects the natural hierarchy between genes and cells, allowing each to influence the other without collapsing their distinct biological roles, enabling HEIST to learn spatially informed, biologically grounded representations that generalize across tissues and proteomics.

Finally, the intra- and cross-level message passing steps are repeated $L$ times, yielding final embeddings

$$\mathbf{Z}_c, \mathbf{Z}_g = \text{HEIST}\left( \mathcal{G}_c(\mathcal{C}, \mathcal{E}, \mathbf{P}, \mathcal{T}), \{\mathcal{G}_g^{t_k}(\mathcal{V}, \mathcal{E}_{t_k}, \mathbf{X}_k)\}_{k=0}^{|\mathcal{C}|} \right).$$

**Decoder.** After calculating the final representations, we pass the embeddings into a decoder to reconstruct the original spatial locations using HEIST-Decoder:

$$\hat{\mathbf{P}}, \{\hat{\mathbf{X}}_k\}_{k=0}^{|\mathcal{C}|} = \text{HEIST-Decoder}\left( \mathcal{G}_c(\mathcal{C}, \mathcal{E}), \{\mathcal{G}_g^{t_k}(\mathcal{V}, \mathcal{E}_{t_k})\}_{k=0}^{|\mathcal{C}|}, \mathbf{Z}_c, \mathbf{Z}_g \right).$$

where, HEIST-Decoder is a 3-layer GIN network (Xu et al., 2018).

## 2.2 PRE-TRAINING TASKS.

**Contrastive Learning.** We use a contrastive objective to learn context-aware representations by bringing similar cells and genes—such as neighboring cells of the same type or co-expressed

genes—closer in embedding space while pushing dissimilar pairs apart. This separates functionally distinct cell populations and gene modules, even when spatially close. Additionally, we introduce cross-level contrastive alignment to ensure consistency between gene and cell representations, so cell embeddings reflect gene expression patterns and gene embeddings incorporate spatial context. The contrastive loss is calculated using the equation below:

$$\ell_{c \leftrightarrow c} = \log \frac{e^{\theta(\mathbf{z}_{c,i}, \mathbf{z}_{c,j})/\tau}}{e^{\theta(\mathbf{z}_{c,i}, \mathbf{z}_{c,j})/\tau} + \sum_{k \in \mathcal{N}_i} e^{\theta(\mathbf{z}_{c,i}, \mathbf{z}_{c,k})/\tau}}, \quad \ell_{g \leftrightarrow g} = \log \frac{e^{\theta(\mathbf{z}_{g,p}^i, \mathbf{z}_{g,q}^j)/\tau}}{e^{\theta(\mathbf{z}_{g,p}^i, \mathbf{z}_{g,q}^j)/\tau} + \sum_{k \in \mathcal{N}_i} e^{\theta(\mathbf{z}_{g,p}^i, \mathbf{z}_{g,r}^k)/\tau}},$$

$$\ell_{c \leftrightarrow g} = \log \frac{e^{\theta(\mathbf{z}_{c,i}, \bar{\mathbf{z}}_g^j)/\tau}}{e^{\theta(\mathbf{z}_{c,i}, \bar{\mathbf{z}}_g^j)/\tau} + \sum_{k \in \mathcal{N}_i} e^{\theta(\mathbf{z}_{c,i}, \bar{\mathbf{z}}_g^k)/\tau}}, \quad \mathcal{L}_{\text{contrastive}} = \sum_{i,j \in \mathcal{P}} \left( \ell_{c \leftrightarrow c} + \ell_{c \leftrightarrow g} \right) + \sum_{(p,q) \in \mathcal{V}^2} \ell_{g \leftrightarrow g}.$$

where $\theta(\cdot, \cdot)$ denotes a similarity function (e.g., cosine similarity), $c \leftrightarrow c$, $g \leftrightarrow g$, and $c \leftrightarrow g$ denotes contrastive loss between cells, genes, and cell-genes respectively, $\tau$ is a temperature parameter that controls the sharpness of the contrastive distribution, and $t_i$ is the cell type label of cell $i$. The set of positive pairs $\mathcal{P} = \{(i,j) \mid t_i = t_j, d(i,j) \leq r\}$, where $d(i,j)$ is the spatial distance between cells $i$ and $j$, and $r$ is a fixed spatial radius. Similarly, the set of negative samples for cell $i$, $\mathcal{N}_i = \{k \mid t_i \neq t_k, d(i,k) \leq r\}$, i.e., cells within radius $r$ that belong to a different cell type.

**Masked-auto encoding.** We also train HEIST with a masked auto-encoding loss to improve reconstruction and robustness. By masking subsets of cell and gene nodes, the model learns to reconstruct gene expression and spatial coordinates from the remaining context, reflecting real-world challenges like dropout and noise in spatial transcriptomics. This encourages the model to infer missing data, generalize across datasets, and use gene signals to recover spatial context and spatial cues to predict gene expression. After reconstructing the spatial locations and gene-expression, the masked auto-encoding loss is calculated using equation below:

$$\mathcal{L}_{\text{mae}} = \text{MSE}(\hat{\mathbf{P}} \odot mask_c, \mathbf{P} \odot mask_c) + \frac{1}{|\mathcal{C}|} \sum_{k=0}^{|\mathcal{C}|} \text{MSE}(\hat{\mathbf{X}}_k \odot mask_g^k, \mathbf{X}_k \odot mask_g^k),$$

where $mask_c$ is the mask over the cell locations, and $mask_g^k$ is the gene-expression mask for cell $k$.
**Final loss.** Contrastive learning structures the latent space to emphasize biological similarities and differences, promoting better separation of cell types and gene programs. In contrast, masked autoencoding ensures that embeddings retain rich information content necessary for reconstructing gene expression and spatial locations. Together, they prevent trivial or collapsed representations and produce embeddings that are both discriminative and information-rich. Hence, the final objective is a weighted sum of the contrastive and autoencoding losses, along with an orthogonality regularization which encourages the embedding dimensions to be decorrelated, promoting diverse and non-redundant representations (Zhang et al., 2021):

$$\mathcal{L} = \sigma(\gamma) \cdot \mathcal{L}_{\text{contrastive}} + (1 - \sigma(\gamma)) \cdot \mathcal{L}_{\text{mae}} + \lambda \left( \left\| \mathbf{I}_d - \mathbf{Z}_c^\top \mathbf{Z}_c \right\|_F^2 + \left\| \mathbf{I}_d - \mathbf{Z}_g^\top \mathbf{Z}_g \right\|_F^2 \right),$$

where $\lambda$ is a regularization weight, $\sigma$ is sigmoid function to balance the loss terms, and $\gamma$ is a learnable scalar that dynamically balances two terms.

**Computational efficiency.** As shown in Table A4 in the Appendix, HEIST demonstrates significant computational advantages, achieving **8× faster** embedding extraction time compared to SCGPT-SPATIAL and **48× faster** than SCFOUNDATION. This efficiency comes from HEIST's sparse modeling, which avoids the expensive full self-attention computations required by transformer based models like SCGPT-SPATIAL.

## 3 EXPERIMENTS

In this section, we first describe the pretraining datasets used to train HEIST. We then outline the downstream tasks and corresponding datasets, followed by baselines, results, insights, and ablation studies.

**Pretraining Datasets.** HEIST is trained on a large and diverse collection of high-resolution spatial transcriptomics datasets, primarily generated using single-cell technologies such as MERFISH and Xenium. The pretraining dataset comprises 22.3M cells from 124 tissue slices across 15 organs [cf. Figure 1(A)], including 13.3M cells from 10x Genomics, 8.7M from Vizgen, and 360K cells from the

Seattle Alzheimer's Brain Atlas. The large scale and diversity of the dataset improve the reliability and robustness of learned representations, enabling better transferability to downstream tasks across varying biological contexts and technologies. A detailed breakdown of datasets, descriptions, and sources is provided in Appendix D, and we provide a spreadsheet of the dataset details in the supplementary material.

**Experimental setup.** We provide pretraining hyperparameters in Appendix Table A1. HEIST was pretrained on 4 NVIDIA L40s GPUs (40GB each), with each epoch taking approximately 3 hours. Although the maximum number of epochs was set to 20, early stopping based on validation loss typically halted training around Epoch 5 or 6. For downstream evaluations, we assess HEIST in both zero-shot and fine-tuning settings. In the zero-shot setting, the pretrained model is directly evaluated on unseen spatial transcriptomics datasets to assess generalization without further training. For fine-tuning, we first extract embeddings from the frozen model and either train an MLP prediction head or fine-tune the decoder. We perform each experiment 5 times provide mean and standard deviation, except in Charville and UPMC datasets where we use the split the data using method from (Wu et al., 2022). The code is available at `https://github.com/Graph-and-Geometric-Learning/HEIST`.

**Downstream Tasks.** We evaluate HEIST across four different spatial transcriptomics and proteomics technologies, five organs, and four downstream tasks—cell clustering, cell type annotation, clinical outcome prediction, and gene imputation—to assess both biological insight discovery and clinical relevance of HEIST.

Cell clustering is critical for discovering novel cell types and understanding how microenvironmental factors shape cellular behavior, particularly in tumor microenvironments. An expressive model should not only cluster cells but also reveal microenvironment-driven subclusters, providing insights into spatially informed cell populations. Clustering is performed using frozen embeddings, and performance is evaluated on datasets two spatial transcriptomics datasets SEA-AD (Gabitto et al., 2024) and Merfish Lung Cancer (Chen et al., 2024), and three proteomics datasets Charville, UPMC, and DFCI (Wu et al., 2022), using normalized mutual information (NMI). Cell type annotation classifies cells into known biological categories, enabling functional interpretation of cellular diversity. HEIST embeddings are extracted from the frozen encoder, and an MLP classifier is trained on labeled datasets including SEA-AD, Charville, UPMC, DFCI, and MERFISH lung cancer, with performance evaluated using F1 score.

Clinical outcome prediction aims to classify entire tissues, predicting outcomes such as immunotherapy response, treatment outcomes, remission status, and placenta condition. This task is essential for clinical decision-making and understanding disease progression. HEIST is evaluated on datasets including proteomics data Charville (Colon), UPMC (Neck), and DFCI (Neck) (Wu et al., 2022) collected using CODEX, skin cancer data (Ptacek et al., 2021) collected using MIBI, and spatial transcriptomics data of placenta collected using Xenium. Predictions are made using frozen cell embeddings and an MLP classifier, and we report AUC-ROC. Gene imputation recovers missing or noisy gene expression values, a common issue in spatial transcriptomics due to measurement limitations. We perform gene imputation by predicting masked gene values, using stratified sampling based on gene sparsity following the approach of Avşar & Pir (2023). This task is evaluated in both zero-shot and by fine-tuning the decoder, reporting Pearson correlation between predicted and true gene expression values. We explain these tasks in further details in Appendix D.2.

**Baselines.** We benchmark HEIST against a diverse set of baselines, including graph-based spatial models, single-cell foundation models, and recent spatial foundation approaches. In particular, we include STAGATE (Dong & Zhang, 2022) and GraphST (Long et al., 2023), which capture local spatial relationships but struggle to generalize across datasets. We also evaluate against scFoundation (Hao et al., 2023), a large-scale single-cell foundation model that ignores spatial context. We also include comparisons with recent spatial foundation models CellPLM (Wen et al., 2023a), NoVAE (Blampey et al., 2024) and scGPT-spatial (Wang et al., 2025), which incorporate spatial information but do not explicitly model hierarchical gene-cell interactions. Finally, for the gene imputation task, we compared against MAGIC a denoising method based on graph diffusion. For proteomics datasets such as UPMC, baselines like scGPT-spatial and CellPLM were originally pretrained on fixed gene vocabularies, supporting only 6 of the 28 markers available. To enable comparison, we report two settings: (i) *unaligned*, where models process only the subset of overlapping genes, and (ii) *aligned*, where we manually remap marker identities to the closest supported genes following prior work. It is important to note that SCGPT pretrains on single-cell spatial trasncritomics data mixed with Visium

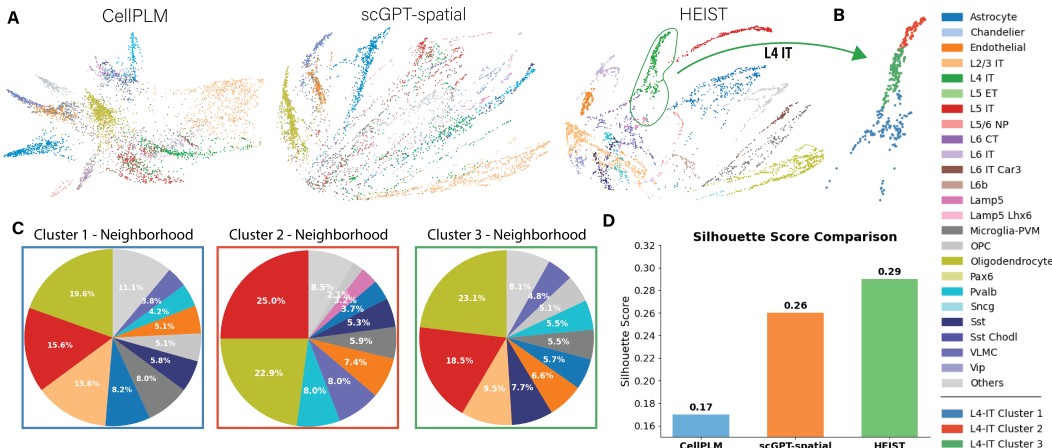

Figure 3: **HEIST accounts for tissue microenvironments.** (A) Comparison of cell embeddings (PHATE) for the same tissue slice from SEA colored by annotated cell types. HEIST demonstrates superior separation compared to other methods. (B) Spectral clustering reveals three *spatially-informed* sub clusters in L4-IT cells. (C) Visualizing the sub clusters neighborhood distribution shows that each cluster accounts for neighborhood differences, demonstrating that HEIST embeddings effectively captures spatial microenvironments through cross-message passing. We show similar results for CEllPLM and scGPT-spatial in Figure A3, showing that these models can not infer spatially-informed clusters. Numerical results are available in Table A3 in Appendix Section E. (D) Silhouette Score comparision between each method.

(spot-level), while CELLPLM combines scRNA-seq with single-cell spatial transcriptomics. HEIST, in contrast, is trained exclusively on single-cell resolution spatial transcriptomics. However, there is major overlap in the single-cell spatial transcriptomics portion of training data between HEIST and the baselines.

## 3.1 RESULTS

**Spatially-aware cell type discovery.** A key strength of HEIST is its ability to uncover *spatially-informed cellular subpopulations*. Figure 3 shows that HEIST embeddings not only separate canonical cell types, but also resolve finer subclusters that align with local microenvironmental context. These subclusters correspond to meaningful biological distinctions. Existing approaches like CELLPLM and SCGPT-SPATIAL fail to capture sub clusters and collapse such substructure, missing microenvironment effects (Figure A3, Appendix). This ability to differentiate spatially-driven heterogeneity is essential for discovering novel biomarkers.

**Gene imputation.** Table 1 reports the performance on the gene imputation task for the placenta and skin datasets. HEIST achieves the best performance after fine-tuning, surpassing all baseline models by **2.5%** on the placenta dataset and **9%** on the skin dataset. We highlight comparisons with MAGIC, as our preprocessing also incorporates this method. The improvements achieved by HEIST can be attributed to its cross-attention mechanism, which in-

Table 1: Performance on gene imputation task

| Model | Placenta | Skin |
|---|---|---|
| MAGIC | $0.749 \pm 0.000$ | $0.671 \pm 0.000$ |
| ScFoundation (Fine-tuned) | $0.721 \pm 0.004$ | $0.621 \pm 0.003$ |
| CellPLM (Fine-tuned) | $\underline{0.801} \pm 0.011$ | $0.723 \pm 0.007$ |
| scGPT-spatial (Fine-tuned) | $0.718 \pm 0.002$ | $\underline{0.740} \pm 0.002$ |
| HEIST (Zero-Shot) | $0.574 \pm 0.000$ | $0.350 \pm 0.000$ |
| HEIST (Fine-tuned) | $\mathbf{0.821} \pm \mathbf{0.041}$ | $\mathbf{0.807} \pm \mathbf{0.020}$ |
| HEIST Imp. % | 2.49 | 9.05 |

tegrates contextual information from neighboring cells to enhance imputation for each target cell. Although HEIST's zero-shot performance is limited by the dataset-specific nature of gene expression patterns, fine-tuning allows the model to adapt effectively by leveraging its hierarchical structure and gene co-expression networks, resulting in improved performance.

**Tissue-classification.** Table 2 reports the tissue-level classification performance of various models across multiple datasets and tasks, measured by AUC-ROC. HEIST consistently outperforms existing models, achieving the highest scores in six out of seven evaluation scenarios. Notably, on the UPMC tasks, HEIST surpasses the foundation models scGPT-spatial and CellPLM by **25.4%** and **30%**, respectively due to scGPT-spatial and CellPLM ignoring most of the available markers. Even after aligning, performance remains limited because these models cannot fully leverage all the available markers. In contrast, HEIST constructs co-expression networks directly from observed proteins,

Table 2: Performance on clinical outcome prediction. We classify cancer outcome, cancer remission, treatment response and placental conditions. Results on the right of vertical are proteomics. NEM stands for "not enough markers" error from Novae when it can not match enough markers. *Results from (Wu et al., 2022)

| Organ | Placenta | Colon | | Neck | | Neck | Skin |
|---|---|---|---|---|---|---|---|
| Dataset | | Charville | | UPMC | | DFCI | Melanoma |
| Task / Model | Condition | Outcome | Recurrence | Outcome | Recurrence | Outcome | Response |
| STAGATE | $0.578 \pm 0.12$ | $0.657 \pm 0.032$ | $0.783 \pm 0.050$ | $0.602 \pm 0.054$ | $0.659 \pm 0.013$ | $0.633 \pm 0.210$ | $0.533 \pm 0.267$ |
| GraphST | $0.659 \pm 0.059$ | $0.828 \pm 0.088$ | $0.645 \pm 0.026$ | $0.582 \pm 0.061$ | $0.683 \pm 0.131$ | $0.567 \pm 0.170$ | $0.644 \pm 0.15$ |
| Space-gm* | - | | 0.793 | 0.696 | **0.863** | 0.883 | 0.873 | - |
| ScFoundation | $0.601 \pm 0.16$ | $0.713 \pm 0.122$ | $0.787 \pm 0.113$ | $0.678 \pm 0.061$ | $0.689 \pm 0.111$ | $0.742 \pm 0.093$ | $0.500 \pm 0.155$ |
| Novae | $0.619 \pm 0.10$ | $0.739 \pm 0.006$ | $0.500 \pm 0.000$ | NEM | NEM | $0.750 \pm 0.095$ | NEM |
| CellPLM (unaligned) | $0.682 \pm 0.15$ | $0.744 \pm 0.006$ | $0.801 \pm 0.032$ | $0.681 \pm 0.134$ | $0.667 \pm 0.045$ | $0.750 \pm 0.083$ | $0.580 \pm 0.133$ |
| CellPLM (aligned) | - | $0.732 \pm 0.007$ | $0.792 \pm 0.030$ | $0.670 \pm 0.128$ | $0.655 \pm 0.043$ | $0.738 \pm 0.080$ | - |
| scGPT-spatial (unaligned) | $0.602 \pm 0.08$ | $0.834 \pm 0.081$ | $0.806 \pm 0.019$ | $0.717 \pm 0.117$ | $0.676 \pm 0.097$ | $0.875 \pm 0.040$ | $0.600 \pm 0.000$ |
| scGPT-spatial (aligned) | – | $0.594 \pm 0.081$ | $0.854 \pm 0.005$ | $0.702 \pm 0.001$ | $0.789 \pm 0.080$ | $0.858 \pm 0.006$ | – |
| HEIST | **0.769 ± 0.06** | **0.861 ± 0.086** | **0.887 ± 0.041** | $0.835 \pm 0.001$ | **0.929 ± 0.030** | **0.937 ± 0.062** | **0.866 ± 0.066** |
| HEIST Imp.% | 12.7 | 3.2 | 3.5 | -3.3 | 5.2 | 7.1 | 44.3 |

Table 3: Performance on cell type annotation. Annotations are provided as a feature in each dataset. Results on the right of vertical are proteomics.

| Organ | Lung | Brain | Colon | Neck | Neck |
|---|---|---|---|---|---|
| Model | | SEA-AD | Charville | UPMC | DFCI |
| STAGATE | $0.2187 \pm 0.0570$ | $0.3304 \pm 0.0625$ | $0.2759 \pm 0.0490$ | $0.0687 \pm 0.0136$ | $0.0685 \pm 0.0213$ |
| GraphST | $0.4081 \pm 0.0658$ | $0.2296 \pm 0.1772$ | $0.3675 \pm 0.0873$ | $0.0617 \pm 0.0199$ | $0.0577 \pm 0.0261$ |
| ScFoundation | $0.150 \pm 0.014$ | $0.2495 \pm 0.1147$ | $0.3220 \pm 0.1421$ | $0.0222 \pm 0.0079$ | $0.041 \pm 0.021$ |
| Novae | NEM | $0.2332 \pm 0.0434$ | $0.2194 \pm 0.0455$ | NEM | $0.0736 \pm 0.0215$ |
| CellPLM (unaligned) | $0.5044 \pm 0.1607$ | $0.6701 \pm 0.0827$ | $0.4760 \pm 0.0669$ | $0.0563 \pm 0.0212$ | $0.0565 \pm 0.0179$ |
| CellPLM (aligned) | - | - | $0.3047 \pm 0.0040$ | $0.0337 \pm 0.0032$ | $0.0413 \pm 0.0015$ |
| scGPT-spatial (unaligned) | **0.5671 ± 0.1685** | $0.5907 \pm 0.0029$ | $0.3494 \pm 0.0624$ | $0.0464 \pm 0.0162$ | $0.0618 \pm 0.0163$ |
| scGPT-spatial (aligned) | – | – | $0.3280 \pm 0.0499$ | $0.2195 \pm 0.0490$ | $0.0953 \pm 0.0190$ |
| HEIST | $0.5126 \pm 0.1170$ | **0.9953 ± 0.0158** | **0.5340 ± 0.1293** | **0.2826 ± 0.0758** | **0.1124 ± 0.0521** |
| HEIST Imp. | -9.6 % | 48.5 | 12.2 | 28.7 | 17.9 |

allowing it to incorporate all available markers without retraining. This structural flexibility underlies its strong generalization to proteomics.

**Cell-type annotation.** Table 3 shows the performance on the cell type annotation task across multiple datasets. HEIST achieves the highest performance in four out of five datasets, with substantial gains in UPMC and DFCI neck datasets (**28.7 %** and **17.9 %** improvements, respectively). Notably, scGPT-spatial was pretrained on the MERFISH Lung Cancer dataset, explaining its strong performance there on that data. These results again highlight the effectiveness and generalizability of HEIST.

**Recovering unique niches based on tissue conditions.** In Figure 4, we visualize the highly attended edges in HEIST's attention blocks. We compare the attention-derived niches across Braak stages, and we can see that a clear progression emerges. In Braak 0, the high-scoring edges form small, isolated neuronal clusters that are relatively sparse and spatially compact. By Braak IV and V, these clusters become more numerous and begin to involve a broader set of cell types, indicating early signs of local microenvironmental restructuring. In Braak VI, the niches become much denser and more spatially extended. This gradual transition from isolated neuronal interactions to larger mixed-cell hubs suggests that HEIST attention captures the increasing microenvironmental complexity and tissue reorganization that accompany the progression of pathology across Braak stages.

**Ligand-receptor pair prediction.** We train a linear probe to predict ligand–receptor (LR) interactions for all edges in the tissue graphs. For each model, we extract the cell embeddings, concatenate the embeddings of the two cells involved in a pair, and pass this concatenated representation through a simple single layer perceptron that predicts whether the pair corresponds to an LR interaction. As shown in the Table A9, HEIST achieves the highest AUC-ROC and outperforms all baselines by a clear margin. In particular, HEIST reaches an AUC-ROC of $0.995 \pm 0.002$. This result combined with the attention based results indicates that its embeddings capture the spatial signatures of LR communication more effectively than scFoundation, CellPLM, and scGPT-spatial.

**Ablations.** Table 4 shows that removing any key component of HEIST leads to performance degradation across tasks, confirming their importance. Hierarchical modeling and spatial information are most critical, with their removal causing the largest drops. Pre-training is crucial, especially

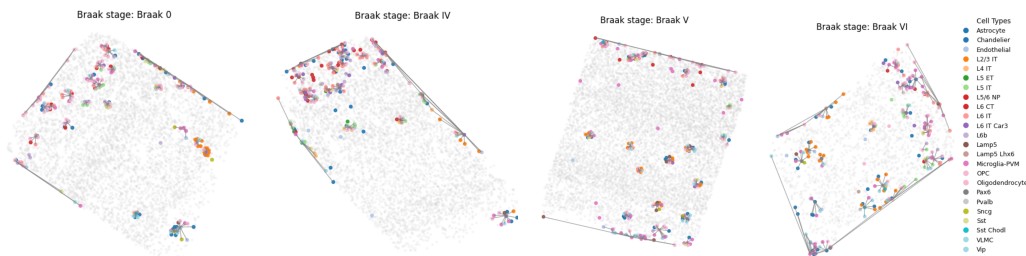

Figure 4: HEIST attention reveals unique microenvironments depending on Braak stages

Table 4: Ablation study showing that hierarchical modeling, cross-level message passing, and the training objectives are critical components for strong performance.

| Model | Charville-Outcome | Skin-Imputation | SEA-Cell classification |
|---|---|---|---|
| HEIST | **0.861 ± 0.086** | **0.807 ± 0.020** | **0.995 ± 0.015** |
| No space (No Hierarchy) | 0.596 ± 0.028 | 0.345 ± 0.010 | 0.179 ± 0.038 |
| No gene (No Hierarchy) | 0.764 ± 0.235 | 0.173 ± 0.014 | 0.194 ± 0.040 |
| No pre-training | 0.500 ± 0.000 | 0.623 ± 0.002 | 0.784 ± 0.128 |
| No cross message passing | 0.625 ± 0.125 | 0.531 ± 0.005 | 0.955 ± 0.041 |
| No positional encodings | 0.523 ± 0.010 | 0.458 ± 0.003 | 0.220 ± 0.034 |
| No contrastive | 0.623 ± 0.002 | 0.536 ± 0.015 | 0.966 ± 0.037 |
| No MAE | 0.658 ± 0.076 | 0.495 ± 0.006 | 0.162 ± 0.038 |
| No orthogonal regularization | 0.594 ± 0.031 | 0.646 ± 0.013 | 0.992 ± 0.020 |

for clinical outcome prediction due to the skewed label distribution in these datasets. Cross-level message passing and contrastive learning significantly improve cell classification, while the MAE is crucial for gene imputation and classification. In Table A5, we compare different positional encoding including Laplacian PEs, random walk based PEs, and sinusoidal PE. We notice that sinusoidal PEs outperform the other PE methods. In Table A6, we present the results with constant and adaptive $\tau$ (threshold for the co-expression graph), showing that the adaptive method we use outperforms the constant threshold. In Table A7, we ablate by removing each of the contrastive term; Results indicate that the three terms combinely capture complementary structural signals. We also present a way to interpret HEIST attention in Figure A4 and Table A8, indicating that HEIST attention focuses on niches and corresponds to known ligand-receptor pairs than existing models. In Table A10, we show show different ways of mixing positional encodings and adding raw features. We notice that passing raw features through MLP before adding them results in performance drop in imputation because because the MLP-based mapping smooths or compresses the raw coordinate and expression signals in a way that removes fine-grained local variation. In Table A11, we show that using smaller batch sizes results in slight decrease in performance as smaller batch sizes provide less microenvironmental cues.

## 4 CONCLUSION

We present HEIST, a hierarchical graph foundation model for spatial transcriptomics that jointly models gene co-expression and spatial adjacency within a unified framework. By integrating biologically motivated hierarchical representation learning with a novel cross-level message passing mechanism, HEIST captures complex dependencies between genes, cells, and tissue-level organization. Pretrained on over 22.3M cells spanning 15 organs, HEIST achieves state-of-the-art performance across multiple downstream tasks and generalizes to proteoomics, while offering significant computational efficiency. Beyond predictive accuracy, HEIST enables the discovery of spatially-informed cellular subpopulations, providing deeper insights into tissue microenvironments. Our results highlight the importance of modeling both molecular and spatial information to advance the development of general-purpose, transferable models for spatial omics.

## ACKNOWLEDGMENTS

S.K. is funded in part by the NIH (NIGMSR01GM135929, R01GM130847), NSF CAREER award IIS-2047856, NSF IIS-2403317, NSF DMS-2327211. S.K is also funded by the Sloan Fellowship

FG-2021-15883, the Novo Nordisk grant GR112933. S.K. and R.Y. acknowledge funding from NSF CISE-2403317. R.Y. has also recieved Amazon Research Award 2024.

## ETHICS STATEMENT

The research presented in this paper fully conforms to the ICLR Code of Ethics. All datasets used in this study are publicly available and were obtained from previously published works with appropriate citations. These datasets were originally collected under IRB approval or equivalent ethical review, with informed consent obtained as documented in the source publications. We only use fully anonymized data and did not conduct any new data collection or direct interactions with human subjects. No personally identifiable information or sensitive patient data is included. Our methods are designed to advance scientific understanding of spatial transcriptomics and biological modeling and do not pose foreseeable risks related to privacy, misuse, or social harm.

## REPRODUCIBILITY STATEMENT

We have taken several steps to facilitate reproducibility of our work. A full description of our model architecture, training objective, pre-training data, hyperparameters, and evaluation metrics is provided in Sections 2-3.1 of the main text, Appendix C.2, D, and supplementary zip. To ensure transparency, we release our anonymized code and configuration files at anonymous repository link, which contains scripts for data preprocessing, training, and evaluation. Furthermore, we provide instructions for reproducing all reported experiments. The code is available at `https://github.com/Graph-and-Geometric-Learning/HEIST`.

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

# A  RELATED WORKS

This section outlines hierarchical graph neural networks and their limitations, followed by graph-based spatial transcriptomics methods and foundation models for single-cell and spatial data.

**Hierarchical graph neural networks.** Hierarchical graph neural networks operate on graphs with multiple levels, where each level represents a distinct granularity or type of information. MSMGN (Fortunato et al., 2022), HOOD (Grigorev et al., 2023), and BSMS-GNN (Cao et al., 2023) perform cross-level message passing across levels, but the levels are of same modality at different resolutions (e.g., coarsened versions of spatial graphs), rather than integrating graphs of distinct modalities and they are geared towards propagating signals globally. HIGH-PPI (Gao et al., 2023) introduces a hierarchical graph where one level encodes protein-protein interactions and the other encodes residue-level interactions. However, it does not include cross-level message passing, which limits the ability to combine information across biological levels. In contrast, HEIST works information from different modalities while integrating information across levels.

**Graph-based Spatial Transcriptomics methods.** Graph-based spatial transcriptomics methods construct cell graphs based on spatial proximity and apply graph neural networks to model spatial relationships. SPAGCN (Hu et al., 2021), CCST (Li et al., 2022), STAGATE (Dong & Zhang, 2022), CONST (Zong et al., 2022), and GRAPHST (Long et al., 2023) train graph neural networks in un/self-supervised manner by using techniques such as deep graph infomax (Veličković et al., 2018), iterative clustering (Braun et al., 2022), and auto-encodings task on gene expression (Fang et al., 2024). Unlike HEIST, these models do not generalize to other tissues without retraining and do not incorporate hierarchical modeling across genes and cells.

**scRNA-seq and Spatial Transcriptomics Foundation models.** Foundation models for single-cell and spatial transcriptomics aim to pretrain general-purpose representations that can be transferred across datasets and tasks. SCFOUNDATION (Hao et al., 2023), GENEFORMER (Theodoris et al., 2023), and SCGPT (Cui et al., 2024) extend transformer architectures to model scRNA-seq data by treating gene expression profiles as sequences. However, these models assume an ordering over genes and do not explicitly capture cell-cell relationships or spatial information. SCGPT-SPATIAL (Wang et al., 2025) adapts SCGPT to spatial transcriptomics. CELLPLM (Wen et al., 2023a) incorporates cell-cell interactions in pertaining and adds a gaussian mixture prior to overcome data limitations. But CELLPLM and SCGPT-SPATIAL operate over a fixed set of genes and do not model hierarchy. While transformer based models like SCGPT-SPATIAL can be considered as graph neural networks over fully conneceted graphs Joshi (2025), they most biological networks are far from being fully connected, and explicitly modeling sparse, biologically plausible interactions is critical and enabled by graph-based learning. Existing foundation models thus either neglect spatial dependencies, gene co-expression structure, or both, limiting their ability to capture the context-specific nature of spatial transcriptomics. Furhtermore, their reliance in on gene embeddings hinders their generalization to proteomics data. In contrast, HEIST hierarchically models both this information and computes expressive and generalizable representations by working over gene co-expression networks.

# B  DATA PREPROCESSING AND GRAPH CREATION

HEIST requires construction of two graph structures: (1) a spatial cell-cell graph capturing local tissue organization, and (2) cell-type-specific graph representing the gene co-expression networks, which captures gene-gene dependencies conditioned on cell types. We present the preprocessing pipeline in Figure A1, and we describe the data preprocessing steps and the detailed procedures for constructing these graphs below.

**Data Preprocessing.** We begin by importing raw spatial transcriptomics data, followed by standard preprocessing steps, including outlier removal, gene expression normalization, and gene filtering to retain highly variable genes. We denote the raw matrix by $X_{\mathrm{raw}} \in \mathbb{R}^{N \times G}$ and the normalized matrix by $X$, with $X_{ij} = \mathrm{Norm}(X_{\mathrm{raw},ij})$. To mitigate technical noise and dropout effects common in single-cell measurements, we apply MAGIC (Dijk et al., 2017) to denoise gene expression data before all downstream computations. Formally,

$$\tilde{X} = \mathrm{MAGIC}(X).$$

**Cell-Cell Graph Construction.** To capture spatial relationships between cells, we construct a cell-cell graph based on physical proximity. Each cell has spatial coordinates $p_i = (x_i, y_i) \in \mathbb{R}^2$. Spatial locations are used to compute Voronoi polygons, which define local neighborhoods. Edges in the graph are assigned based on adjacency in the Voronoi diagram, ensuring that the graph structure accurately reflects tissue architecture and local microenvironments. We write this as

$$(i, j) \in \mathcal{E}_c \iff \text{Vor}(p_i) \text{ and } \text{Vor}(p_j) \text{ share a boundary.}$$

**Cell-Type-Specific Gene Co-Expression Networks Construction.** For each cell type, we build a gene co-expression network that captures functional dependencies between genes. If cell type annotations are provided, we use them directly; otherwise, we infer cell types via Leiden clustering on the denoised gene expression data. This is expressed as $t_i = \text{Leiden}(\tilde{X})$.

After cell types are assigned, we compute pairwise mutual information (MI) between genes within each cell type using the denoised expression values. Gene pairs with MI greater than a threshold $\tau$ are connected by an edge in the gene co-expression network, resulting in a sparse, cell-type-specific co-expression graph. This process captures gene-gene co-expression patterns and co-expression dependencies unique to each cellular context. For a cell type $t_k$ with cell index set $\mathcal{I}_k = \{i : t_i = t_k\}$,

$$\text{MI}_{ab}^{(k)} = I\left(\tilde{X}_{\cdot,a}^{(k)}; \tilde{X}_{\cdot,b}^{(k)}\right), \qquad (a, b) \in \mathcal{E}_g^{(k)} \iff \text{MI}_{ab}^{(k)} \geq \tau_k.$$

**Choice of $\tau$.** The threshold $\tau$ used for constructing the gene co-expression networks is data-driven and cell-type-specific. Specifically,

$$\tau_k = \mu_k + \sigma_k, \qquad \mu_k = \mathbb{E}_{a \neq b}[\text{MI}_{ab}^{(k)}], \qquad \sigma_k = \text{Std}\left(\{\text{MI}_{ab}^{(k)}\}_{a \neq b}\right).$$

This adaptive choice yields graphs with 20–30% edge density, which empirically provides sufficient structure to capture co-expression dynamics without over-saturating the graph. In earlier development, we experimented with a constant global $\tau$, but this produced artifacts: some cell types became overly dense while others became nearly edge-free. This mismatch harms hierarchical message passing and leads to degraded downstream performance. We also noticed similar trends in graph based baselines such as STAGATE, GraphST, and Novae, where we had to tune the thresholds manually to control the connectivity. We have highlighted these details in Appendix Section B.

**Batching the tissue.** Each tissue is divided into spatially coherent blocks that contain only a subset of nearby cells. Let $B$ denote the block size and $\{\mathcal{B}_1, \ldots, \mathcal{B}_M\}$ be the resulting partition with $|\mathcal{B}_m| \leq B$. In practice, we choose the largest block size that fits within the memory budget of a single GPU; for our hardware (40GB L40), this is 256 cells per block. Larger blocks allow HEIST to capture mid-range spatial interactions more effectively, while still avoiding the quadratic cost of full-tissue attention. We expect diminishing returns beyond a certain block size: once the local microenvironment and its spatial context are fully captured, adding more distant cells contributes little additional signal.

**Final Output.** The final preprocessing pipeline produces:

- A spatial $\mathcal{G}_c(\mathcal{C}, \mathcal{E}, \mathbf{P}, \mathcal{T})$ **cell-cell graph** encoding local tissue structure.
- **Cell-type-specific gene co-expression networks** $\{\mathcal{G}_g^{t_k}(\mathcal{V}, \mathcal{E}_{t_k}, \mathbf{X}_k)\}_{k=0}^{|\mathcal{C}|}$ capturing co-expression function within each cell type.

These graph structures provide the foundation for HEIST's hierarchical learning framework, enabling integration of spatial organization and gene co-expression patterns.

**Final Output.** The final preprocessing pipeline produces:

- A spatial $\mathcal{G}_c(\mathcal{C}, \mathcal{E}_c, P)$ **cell-cell graph** encoding local tissue structure.
- **Cell-type-specific gene co-expression networks** $\{\mathcal{G}_g^{t_k}(\mathcal{V}, \mathcal{E}_{t_k}, X_k)\}_{k=0}^{|\mathcal{C}|}$ capturing co-expression function within each cell type.

These graph structures provide the foundation for HEIST's hierarchical learning framework, enabling integration of spatial organization and gene co-expression patterns.

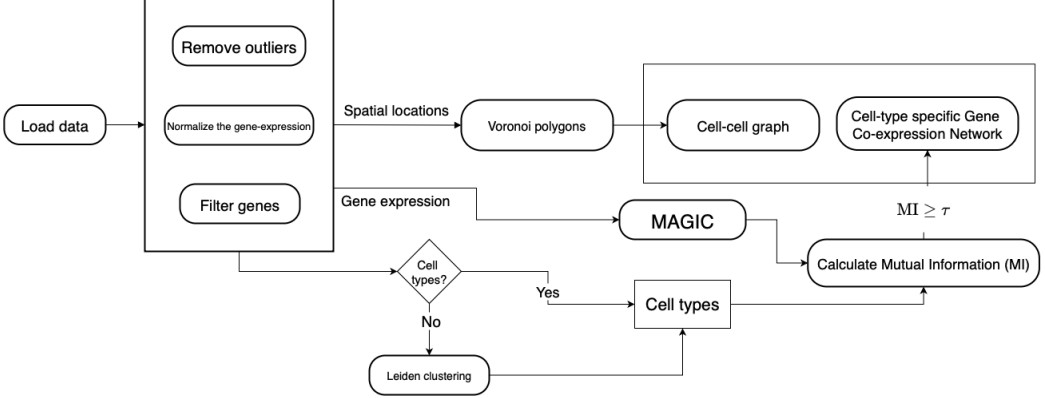

Figure A1: HEIST preprocessing pipeline

## C   IMPLEMENTATION DETAILS

In this section, we explain the implementation details including batching, the positional encoding initialization, graph transformer architecture, and hyperparameters used in pre-training phase.

### C.1   POSITIONAL ENCODINGS

HEIST incorporates positional encodings (PE) at both the cell and gene levels to inject spatial and co-expression structure into the learned representations. As opposed to traditional graph PEs such has Laplacian PE (Dwivedi & Bresson, 2020), random walk PE (Dwivedi & Bresson, 2020), node-centrality based PE (Ying et al., 2021), we make use of sinusoidal PEs. This design choice is motivated by the fact that sinusoidal PE yields expressive representations and are computationally in-expensive as opposed to traditional graph PEs, leading to efficient and expressive representations (Wen et al., 2023b). Below, we describe how these encodings are constructed and why they are suitable for spatial transcriptomics.

Each cell is associated with a spatial coordinate $(x, y) \in \mathbb{R}^2$, corresponding to its location in the tissue. To encode spatial information, we apply a two-dimensional extension of the sinusoidal positional encoding introduced in transformers (Vaswani et al., 2017):

$$\mathrm{PE}_{c,2i} = \sin\left(\frac{x}{10000^{4i/d}}\right), \quad \mathrm{PE}_{c,2i+1} = \cos\left(\frac{x}{10000^{4i/d}}\right),$$
$$\mathrm{PE}_{c,2j+d/2} = \sin\left(\frac{y}{10000^{4j/d}}\right), \quad \mathrm{PE}_{c,2j+1+d/2} = \cos\left(\frac{y}{10000^{4j/d}}\right),$$

where $d$ is the dimensionality of the encoding. To calculate the gene-PE, for each cell, genes are first sorted in descending order of expression. Since most of the spatial transcriptomics are noisy, ranking is done after denoising using MAGIC. The rank of each gene $g$ reflects its relative expression within the cell:

$$\mathrm{Rank}_k(g) = \text{position of } g \text{ in sorted list by expression of cell } k.$$

We then apply sinusoidal encoding based on the rank values as follows:

$$\mathrm{PE}_{g,2i}^k = \sin\left(\frac{\mathrm{Rank}_k(g)}{10000^{2i/d}}\right), \quad \mathrm{PE}_{g,2i+1}^k = \cos\left(\frac{\mathrm{Rank}_k(g)}{10000^{2i/d}}\right).$$

This treats genes within a cell as a soft sequence, allowing the model to distinguish highly expressed genes from lowly expressed ones based on their position in the transcriptional program. Sinusoidal encodings on rank preserve relative ordering and allow the model to capture patterns in gene expression. Since the same gene can have different ranks across cells, the resulting embedding is context-dependent, enabling the model to learn cell-specific co-expression roles of genes. This is important for spatial transcriptomics, where the function and relevance of a gene may vary depending on the cell type and microenvironment.

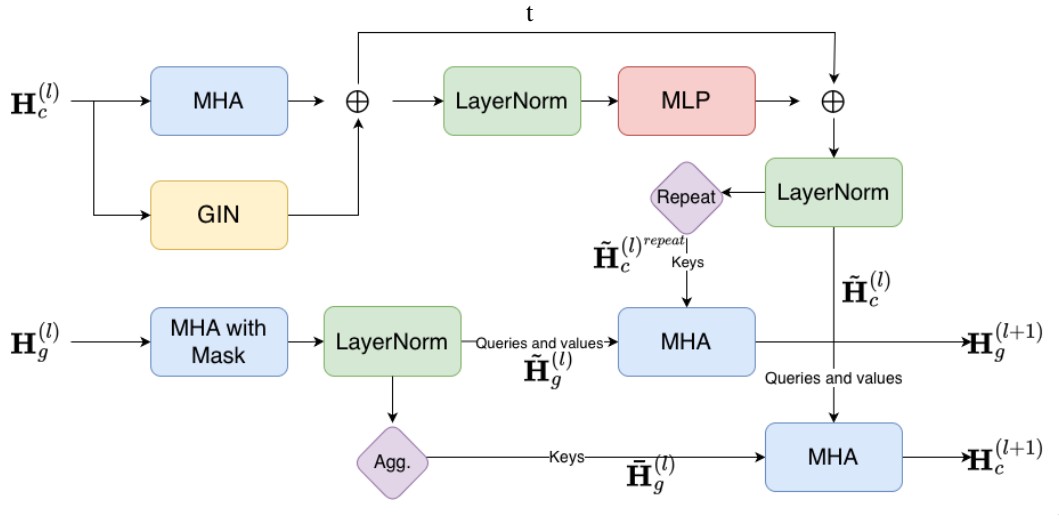

Figure A2: HEISTLayer Block

$$\text{PE}_c = \text{2DSinusoidal}(\mathbf{P}), \text{PE}_g^k = \text{RankSinusoidal}(\mathbf{X}_k); \quad \mathbf{H}_c^{(0)} = \text{PE}_c + \mathbf{P}_c^{\text{repeat}}, \mathbf{H}_g^{k^{(0)}} = \text{PE}_g^k + \mathbf{X}_k$$

where $\mathbf{H}_c^{(0)}$ and $\mathbf{H}_g^{k^{(0)}}$ are the input cell and gene embeddings for cell $k$, respectively. Since we have 2D sinusoidal PEs, the first $\frac{d}{2}$ half of the dimensions correspond to the $x$ co-ordinate from the $(x, y)$ pair. To add $(x, y)$ pair to the $d$ dimensional vector, we repeat the co-ordinates as $[x, \dots \frac{d}{2}$ times, $x, y, \dots \frac{d}{2}$ times, $y]$. This way we are adding two $d$ dimensional vectors. Similarly for genes, we add the scalar value of gene expression to each dimension. Together, the PE helps HEIST incorporate spatial geometry and transcriptional ordering in a biologically meaningful and computation-efficient way.

## C.2 INTRA-LEVEL MESSAGE PASSING

We incorporate biological inductive biases by applying global (all-to-all) attention on the cell graph to capture long-range spatial interactions and local attention that depends on the graph structure of gene graphs to model fine-grained co-expression patterns (Papalexi & Satija, 2018). Capturing global dependencies is crucial for detecting large-scale tissue structures, such as immune infiltration zones and tumor-stroma boundaries (Zhu et al., 2024), while modeling local co-expression networks enables identification of tightly coordinated gene modules within specific niches (Halpern et al., 2017). HEIST performs intra-level message passing independently at the cell and gene levels to capture both local and global dependencies within each modality.

At the *cell level*, we compute:

$$\tilde{\mathbf{H}}_c^{(l)} = \text{GraphTransformerLayer}(\mathbf{H}_c^{(l-1)}) = \text{TransformerLayer}(\mathbf{H}_c^{(l-1)}) + \text{GIN}(\mathbf{A}_c, \mathbf{H}_c^{(l-1)}),$$

where $\mathbf{H}_c^{(l)}$ denotes the cell embeddings at layer $l$, and $\mathbf{A}_c$ is the adjacency matrix of the spatial cell graph. The transformer layer enables long-range interactions across spatial regions, while the GIN layer aggregates local neighborhood information. This hybrid approach captures both global spatial context (e.g., tissue-level structure) and localized microenvironments (e.g., niches or boundaries). Although the attention module is fully dense within each block, the computational cost is significantly reduced because the effective sequence length becomes the block size rather than the total number of cells in the tissue. A naive full-tissue attention operation would scale as $O(c^2)$, which is expensive for large tissues. After spatial batching, the complexity becomes $\mathcal{O}(b^2)$, where $b$ is the block size. In practice, $b$ is much smaller than the total number of cells in a tissue, so this reduction makes attention computation tractable while still capturing local and mid-range spatial interactions.

Table A1: Hyperparameters and their default values used in the experiments.

| Hyperparameter | Default Value |
| --- | --- |
| Positional encoding dimension | 128 |
| Hidden dimension | 128 |
| Output dimension | 128 |
| Number of `HEISTLayer` layers | 10 |
| Number of transformer heads | 8 |
| Batch size | 256 |
| Learning rate | 0.001 |
| Weight decay | 0.003 |
| Number of pre-training epochs | 20 |
| Activation function | GeLu (Hendrycks & Gimpel, 2016) |
| Optimizer | AdamW (Loshchilov & Hutter, 2017) |

At the *gene level*, we apply sparse attention based on the connectivity of the genes as:

$$\alpha_{ij}^{k^{(l)}} = \frac{(\mathbf{W}_q \mathbf{h}_{g,i}^{k^{(l-1)}})^\top (\mathbf{W}_k \mathbf{h}_{g,j}^{k^{(l-1)}})}{\sqrt{d}}, \quad \text{if } (i,j) \in \mathcal{E}_k,$$

$$\tilde{\mathbf{h}}_{g,i}^{k^{(l)}} = \sum_{j \in \mathcal{N}_g^k(i)} \text{softmax}_j(\alpha_{ij}^{k^{(l)}}) \mathbf{W}_v \mathbf{h}_{g,j}^{k^{(l-1)}},$$

where $\mathbf{h}_{g,i}^{k^{(l)}}$ is the embedding of gene $i$ in cell $k$ at layer $l$, and $\mathcal{N}_g^k(i)$ denotes the gene coexpression network neighbors of gene $i$ in that cell. This formulation ensures attention is computed only over biologically relevant interactions, preserving co-expression sparsity while allowing expressive, cell-specific dependency modeling.

### C.3 HYPERPARAMETERS

In Table A1, we describe the hyperparameters for the pre-training model.

## D DATASETS

In this section, we first briefly talk about pre-training datasets, followed by downstream tasks explained in details and downstream datasets.

### D.1 PRETRAINING DATASET

The pre-training dataset of HEIST comprised of 22.3 million cells from 124 tissues slices and 15 organs, including data collected through MERFISH and Xenium. The data consists of 13.3M cells from 10xGenomics datasets (https://www.10xgenomics.com/), 8.7M from Vizgen (https://vizgen.com/) and 360k cells from Seattle Alzheimer's Brain Atlas (https://portal.brain-map.org/explore/seattle-alzheimers-disease).

In contrast to previous models that incorporate lower-resolution data from Visium arrays (Cui et al., 2024), HEIST focuses exclusively on single-cell resolution datasets. This design choice reflects the fact that Visium data lacks true single-cell resolution, instead capturing transcriptomic signals aggregated over spatial spots that often contain heterogeneous mixtures of multiple cell types. Such coarse-resolution data can introduce confounding signals and limit the model's ability to accurately learn cell-level spatial dependencies and gene co-expression. By focusing on single-cell spatial transcriptomics, HEIST is better positioned to capture fine-grained spatial organization, cell-cell interactions, and context-dependent gene co-expreesion critical for modeling tissue microenvironments and cellular heterogeneity.

### D.2 DOWNSTREAM TASKS

In this section, we explain the tasks in more details

**Cell type annotation.** This task involves assigning cells to known cell types based on their gene expression and spatial context. We evaluate this by extracting representations from a frozen HEIST

Table A2: Summary of spatial-omics datasets used in this study. For each dataset, we indicate the organ of origin, number of tissue slices analyzed, total cell count, imaging technology used, and the tasks performed.

| Dataset | Organ | # Slices | # Cells | Technology | Tissue Classification | Cell-type Annotation | Clustering | Gene imputation |
|---------|-------|----------|---------|------------|----------------------|---------------------|------------|-----------------|
| Lung Cancer | Lung | 1 | 100,000 | MERFISH | × | ✓ | ✓ | × |
| DFCI | Head and neck | 58 | 125,512 | CODEX | ✓ | ✓ | ✓ | × |
| UPMC | Head and neck | 308 | 2,164,932 | CODEX | ✓ | ✓ | ✓ | × |
| Charville | Colon | 292 | 632,180 | CODEX | ✓ | ✓ | ✓ | × |
| Melanoma | Skin | 54 | 540,000 | MIBI | ✓ | × | × | ✓ |
| Placenta | Placenta | 212 | 1,000,000 | Xenium | ✓ | × | × | ✓ |

encoder and training only an MLP head for classification. Accurate cell type annotation is essential for characterizing cellular composition across tissues and studying how cell populations contribute to tissue function and disease.

**Cell clustering.** Cell clustering is performed in an unsupervised manner using frozen HEIST embeddings to discover both known and novel cellular subpopulations. Unlike prior models, HEIST's spatially informed representations not only separate cells into canonical cell types but also induce *subclusters* that reflect the influence of local microenvironments. This enables the discovery of previously unrecognized, spatially contextualized cell types based on their tissue niche and surrounding cellular interactions.

**Tissue-level classification.** In this task, the goal is to predict tissue-level outcomes, including response to immunotherapy, treatment outcomes, remission status, and placenta condition. Accurate classification requires capturing both local cellular microenvironments and broader tissue organization patterns. We first evaluate HEIST on datasets from Charville (Colon), UPMC (Neck), and DFCI (Neck), collected using CO-Detection by Indexing (CODEX) technology. We also predict immunotherapy response on a skin cancer dataset (Ptacek et al., 2021) collected through multiplex ion beam imaging (MIBI). Finally, we classify placenta condition into normal placenta, placenta accreta spectrum (PAS), and placental insufficiency using data collected with Xenium. Spatial modeling is crucial for placenta classification, as conditions like PAS involve disrupted tissue architecture and abnormal cell invasion, which gene expression alone cannot fully capture. For this task, we extract cell representations from a frozen HEIST encoder, we aggregate the cell embeddings by taking a mean, and train an MLP head. We report AUC-ROC as the evaluation metric.

**Gene imputation.** In spatial transcriptomics, gene expression profiles are often incomplete or noisy due to technical limitations and measurement dropouts. Gene imputation aims to predict missing gene expression values, recovering biologically plausible expression patterns. HEIST performs this task by leveraging the learned gene embeddings, which capture both co-expression relationships from gene co-expression networks and spatial dependencies from their parent cells. Through cross-level message passing, the model can incorporate spatial cues to refine gene predictions, enabling more accurate reconstruction of missing data. We select the genes to be masked using stratified sampling according to gene sparsity (Avşar & Pir, 2023) and mask these genes during the fine-tuning phase. For the skin dataset, which contains 30 proteomic markers, we masked 4 genes using stratified sampling based on gene sparsity following Avsar et al., corresponding to a 14% masking fraction. For the placenta dataset, we masked 16 out of 200 genes ( 8% masking). This task can be performed in a zero-shot fashion or by fine-tuning the HEIST decoder on the downstream task. We report Pearson correlation as the evaluation metric for this task.

### D.3 DOWNSTREAM DATASETS

We have curated different datasets for each downstream prediction task. To show our models generalize well we specifically used 4 different technologies across the different tasks. The dataset description can be found on Table A2.

## E ADDITIONAL RESULTS

As shown in Figure 3, HEIST effectively identifies such microenvironment-specific subpopulations, which existing foundation models fail to capture. Figure A3 does the same analysis on scGPT-spatial and CellPLM showing that these methods fail to address the microenvironment separation.

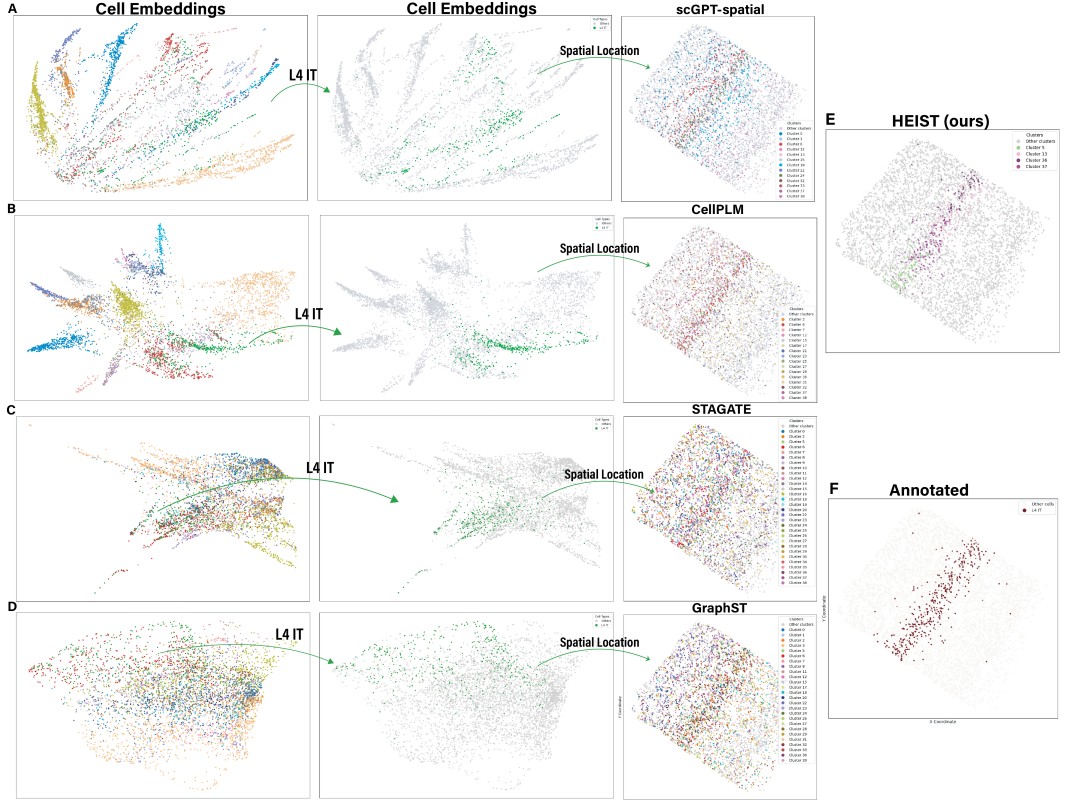

Figure A3: Comparisons between embeddings focusing on L4-IT neurons. The figure shows the cell type embeddings, followed by spectral clustering and a plot of all clusters that contain the specific cell type. **A.** scGPT-spatial results. **B.** CellPLM results **C.** STAGATE results **D.** GraphST results **E.** HEIST results (same as Figure 3) **F.** Ground Truth cell types colored in space.

Table A3: Comparison of unsupervised cell clustering performance using Normalized Mutual Information (NMI).

| Organ | Brain | Colon | Neck | Neck | Lung |
|---|---|---|---|---|---|
| Model | SEA-AD | Charville | UPMC | DFCI | MERFISH Lung cancer |
| STAGATE | $0.294 \pm 0.05$ | $0.237 \pm 0.04$ | $0.022 \pm 0.01$ | $0.048 \pm 0.02$ | $0.171 \pm 0.06$ |
| GraphST | $0.444 \pm 0.02$ | $0.224 \pm 0.05$ | $0.020 \pm 0.01$ | $0.044 \pm 0.02$ | $0.297 \pm 0.05$ |
| ScFoundation | $0.388 \pm 0.04$ | $0.220 \pm 0.09$ | $0.020 \pm 0.014$ | $0.041 \pm 0.02$ | $0.049 \pm 0.00$ |
| CellPLM | $0.651 \pm 0.00$ | $0.24 \pm 0.00$ | $0.015 \pm 0.00$ | $0.075 \pm 0.00$ | $0.286 \pm 0.00$ |
| scGPT-spatial (aligned) | $0.674 \pm 0.04$ | $0.253 \pm 0.07$ | $0.017 \pm 0.01$ | $0.038 \pm 0.01$ | $\mathbf{0.307 \pm 0.08}$ |
| HEIST | $\mathbf{0.691 \pm 0.04}$ | $\mathbf{0.297 \pm 0.03}$ | $\mathbf{0.043 \pm 0.01}$ | $\mathbf{0.14 \pm 0.04}$ | $0.274 \pm 0.04$ |

**Cell clustering.** Table A3 reports normalized mutual information (NMI) scores for the cell clustering task across multiple organs and datasets. HEIST achieves the highest or comparable NMI scores in most settings, demonstrating its ability to capture meaningful cellular subpopulations informed by both spatial and co-expression. Notably, HEIST shows substantial improvement on datasets with complex tissue structures, such as UPMC and DFCI, where modeling spatial microenvironments is critical. While some models achieve competitive performance in simpler settings, they fail to generalize as effectively across technologies due to missing genes in their gene set.

**Runtime comparison.** Table A4 compares the runtime efficiency of HEIST with existing foundation models for generating embeddings from a tissue sample with 19,826 cells. HEIST achieves substantial speed improvements, being **8× faster** than SCGPT and **48× faster** than SCFOUNDATION, while providing both cell and gene embeddings. We also present show the coputational complexity analysis of existing models. We note that HEIST is linear in gene edges and gene count, one HEISTLayer is faster than one scGPT-spatial and scFoundation layer, however it is slower than CellPLM. Although

Table A4: Runtime comparison for embedding extraction from a tissue with 19826 cells, and computational complexity of each model. *CellPLM does not compute gene representations.

| Model | Runtime (s) | Complexity | Interpretation |
|---|---|---|---|
| scFoundation | 290.48 | $\mathcal{O}(C^2 G_{\text{nz}}^2 d + CTd)$ | Quadratic in cell count and nonzero genes. Decoder is linear due to Performer attention. |
| scGPT | 49.37 | $\mathcal{O}(T^2 d + Td^2)$ | Quadratic in cell times number of gene tokens per spot |
| HEIST | **6.69** | $\mathcal{O}\big(d(C^2 + E_c + CE_g + CT)\big)$ | Quadratic in cells (global spatial transformer), linear in gene edges and gene count. |
| CellPLM* | 1.97 | $\mathcal{O}(C^2 d)$ | Quadratic in cells, linear in hidden size. No dependence on number of genes after aggregation. |

Table A5: Ablation comparing sinusoidal PE based on spatial coordinates with traditional graph-based PEs for clinical outcome prediction.

| Organ | Colon | | Neck | | Neck |
|---|---|---|---|---|---|
| Dataset | Charville | | UPMC | | DFCI |
| Task
Model | Outcome | Recurrence | Outcome | Recurrence | Outcome |
| Random walk PE | $0.770 \pm 0.034$ | $0.695 \pm 0.042$ | $0.776 \pm 0.029$ | $0.752 \pm 0.038$ | $0.916 \pm 0.026$ |
| Laplacian PE | $0.750 \pm 0.031$ | $0.800 \pm 0.025$ | $0.670 \pm 0.045$ | $0.810 \pm 0.021$ | $0.852 \pm 0.040$ |
| Sinusoidal PE | $\mathbf{0.861 \pm 0.086}$ | $\mathbf{0.887 \pm 0.041}$ | $\mathbf{0.835 \pm 0.001}$ | $\mathbf{0.929 \pm 0.030}$ | $\mathbf{0.937 \pm 0.062}$ |

CellPLM is faster, it does not compute gene representations, making it less suitable for tasks requiring joint cell and gene analysis.

**PE ablation.** Table A5 presents an ablation study comparing sinusoidal PE against traditional graph-based PEs—random walk and Laplacian—on clinical outcome prediction task. Sinusoidal PE consistently yields higher accuracy and lower variance, particularly excelling in recurrence prediction, highlighting its effectiveness in capturing spatial patterns relevant to clinical signals.

$\tau$ **choice ablation.** In Table A6, we compare the results of tissue classification between adaptive and constant threshold to create gene co-expression graphs. As we can see, the adaptive $\tau$ results consistently outperforms the constant version, capturing fine-grained co-expression structure.

**Importance of contrastive terms.** To understand whether the three contrastive components interact positively, we carried out an ablation experiment in which we removed each term in turn: cell to cell, gene to gene, and cell to gene. As shown in Table A7, removing any single term consistently reduces performance across all datasets. Removing the cell to cell objective reduces Charville outcome prediction from 0.861 to 0.380. Removing the gene to gene objective produces the largest reduction on SEA cell classification, which drops from 0.995 to 0.610. These results show that each contrastive component captures complementary structural signals that include local cell similarity, gene program structure, and cross scale coupling between cells and genes. Using all three terms together provides the strongest supervision signal and does not create negative interactions. We also tested with a joint contrastive loss function that directly models the c–c, g–g, and c–g correlations together rather than simply adding the three contrastive losses. As we can see, the performance with this joint contrastive loss decreased slightly. The primary reason is that the joint contrastive objective significantly increased memory consumption, causing repeated OOM errors. To stabilize training, we were forced to reduce the batch size. Smaller batches provide far less microenvironmental diversity within each step, which is essential for learning strong contrastive signals. This reduced context ultimately impaired the effectiveness of the joint contrastive loss and led to the observed decrease in performance.

**Attention interpretation.** We analyzed the cell to cell attention scores learned by HEIST and CellPLM and visualized their top attention edges. We aggregated the attention across layers and heads to get a single attention maps. We only compared with CellPLM since that is the only other foundation model that calculates cell-cell attention.

Table A6: Ablation comparing constant and adaptive choice of $\tau$.

| Dataset | Charville | Charville | UPMC | UPMC | DFCI |
| Task $\tau$ | Outcome | Recurrence | Outcome | Recurrence | Outcome |
|---|---|---|---|---|---|
| Constant | 0.68 | 0.70 | 0.828 | 0.75 | 0.916 |
| Adaptive | **0.86** | **0.88** | **0.835** | **0.92** | **0.937** |

Table A7: Ablation of HEIST's contrastive loss components across three tasks. Removing any of the three contrastive terms degrades performance across modalities, confirming that the full joint loss is necessary.

| Loss | Charville-Outcome | Skin-Imputation | SEA-Cell classification |
|---|---|---|---|
| Full contrastive | $0.861 \pm 0.086$ | $0.807 \pm 0.020$ | $0.995 \pm 0.015$ |
| Joint contrastive | $0.808 \pm 0.040$ | $0.775 \pm 0.017$ | $0.994 \pm 0.004$ |
| No $c \leftrightarrow c$ | $0.380 \pm 0.119$ | $0.570 \pm 0.060$ | $0.993 \pm 0.006$ |
| No $g \leftrightarrow g$ | $0.575 \pm 0.049$ | $0.477 \pm 0.033$ | $0.610 \pm 0.097$ |
| No $g \leftrightarrow c$ | $0.559 \pm 0.065$ | $0.693 \pm 0.057$ | $0.928 \pm 0.002$ |

As shown in Figure A4, the edges with the highest attention values in HEIST form clear, spatially coherent microenvironments. We observe groups of cells of different types connected in structured patterns, often resembling local niches such. These visual patterns indicate that HEIST's attention mechanism focuses on meaningful neighborhoods rather than isolated pairs. In contrast, the top-attention edges from CellPLM are diffuse and scattered across the tissue. The highlighted edges do not show consistent spatial structure or cell-type organization. This suggests that CellPLM attention is much noisier and does not preferentially emphasize biologically relevant areas.

To quantify whether these visual differences reflect underlying biological communication, we compared all edges to curated ligand-receptor (LR) interactions from CellPhoneDB. Across the entire dataset, approximately 16 percent of edges correspond to known LR pairs. Randomly sampling 1000 edges also gives about 16 percent, which serves as the expected background rate. As shown in Table A8

HEIST shows strong enrichment for LR edges:

- The top 100 high-attention edges contain 31 percent LR interactions.
- The top 1000 edges contain 27.9 percent LR interactions.
- This is better than chance which yields about 16 percent expected LR pairs.

CellPLM shows the opposite trend. Only 11 percent of its top 100 edges and 8.8 percent of its top 1000 edges correspond to LR interactions. This is clearly below the background rate, which indicates that CellPLM attention values do not align with known biological communication.

**PE initialization.** Our choice to incorporate spatial coordinates and gene-expression scalars by repeating/broadcasting them into the embedding dimension follows the standard additive formulation widely used across transformers. Because the raw spatial coordinates $(x, y)$ and gene expression values are low-dimensional, a direct addition to the $d$-dimensional positional embeddings requires a map into $\mathbb{R}^d$. Repetition provides a simple, parameter-free way to match dimensionality while maintaining the relative scale and directional structure of the original signals. Importantly, we show that initialization that relies purely on an MLP ("No PE" in Table 4). In this formulation, we pass the spatial co-ordinates and gene expression into the latent space using an MLP. This leads to a marked drop in performance, indicating that the positional encodings play an essential role. In Table A10, we ran an additional pretraining run that combines sinusoidal PEs with an MLP-based mapping of coordinates and expression values.

**Batch size ablation.** To evaluate how smaller block sizes affect performance, we ran a pre-training run with the HEIST pipeline using smaller block sizes. We see a slight decrease in the results except for cell classification where the results became almost perfect. This is because batches would provide less microenvironmental cues in each step, leading to the slight drop in performance.

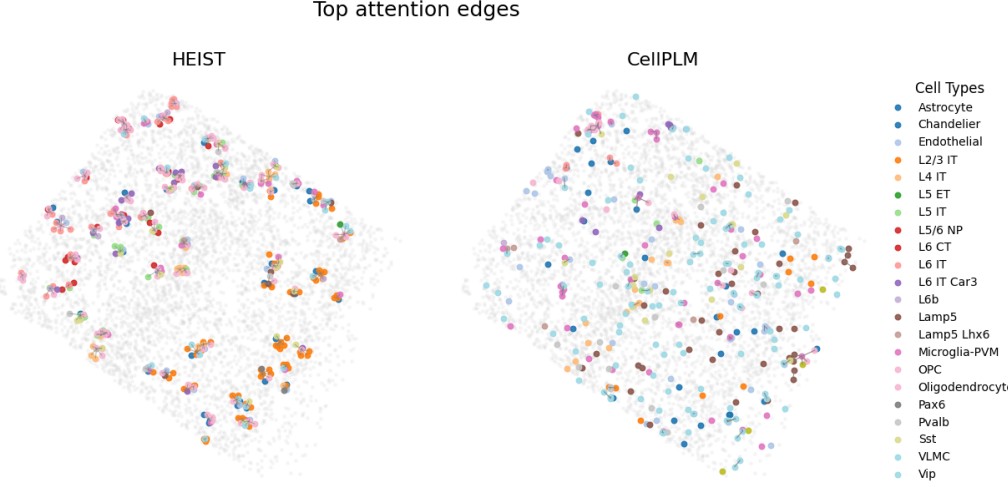

Figure A4: Visualizing edges with highest cell-cell attentions. We can clearly see that HEIST is able to form niches, while CellPLM high attention is scattered around the tissue.

Table A8: LR-rate measured using cell-cell attention.

| Model | LR pairs in top 100 | LR pairs in top 1000 |
|---|---|---|
| Background (all edges) | 0.165 | 0.165 |
| Random | 0.164 | 0.165 |
| CellPLM | 0.110 | 0.088 |
| **HEIST** | **0.310** | **0.279** |

## F    LIMITATIONS

While HEIST advances the state of foundation models for spatial transcriptomics (ST), it has several limitations. First, the current gene co-expression network construction relies on co-expression relationships using mutual information, which may not fully capture causal gene co-expression mechanisms or directional influences. This can be integrated by more sophisticated MI meassures like DREMI (Krishnaswamy et al., 2014). Furthermore, integrating more sophisticated gene co-expression network inference techniques could improve the biological interpretability and efficiency of gene embeddings. A potential direction for future work can be incorporating temporal dynamics by applying techniques such as Granger causality (Tong et al., 2023) over inferred pseudotime trajectories, allowing the model to capture not only static spatial and co-expression dependencies but also directional gene co-expression influence and developmental progression, which are currently not modeled in HEIST. Second, the model assumes static spatial snapshots of tissues and does not account for temporal dynamics or developmental trajectories, which are critical in understanding certain biological processes. Extending HEIST to model spatio-temporal transcriptomics data is an important direction for future work. Finally, while HEIST improves computational efficiency over prior foundation models, it still requires substantial computational resources for large-scale pretraining. Despite these limitations, HEIST provides a flexible and scalable foundation for modeling complex spatial and molecular interactions, and future work can address these challenges to further improve its generalization and interpretability.

Table A9: Predicting LR pairs

| Model | AUC-ROC |
|---|---|
| scFoundation | $0.963 \pm 0.004$ |
| CellPLM | $0.930 \pm 0.006$ |
| scGPT-spatial | $0.984 \pm 0.003$ |
| **HEIST** | $\mathbf{0.995 \pm 0.002}$ |

Table A10: PE initialization ablation.

| PEs | Charville-Outcome | Skin-Imputation | SEA-Cell classification |
|---|---|---|---|
| No PE (MLP($\mathbf{P}$)) | $0.523 \pm 0.010$ | $0.458 \pm 0.003$ | $0.220 \pm 0.034$ |
| Sinusoidal PE (MLP($\mathbf{P}$) + $PE_c$) | $0.857 \pm 0.045$ | $0.538 \pm 0.026$ | $\mathbf{0.999 \pm 0.001}$ |
| HEIST PE ($PE_c + \mathbf{P}_c^{\text{repeat}}$) | $\mathbf{0.861 \pm 0.086}$ | $\mathbf{0.807 \pm 0.020}$ | $0.995 \pm 0.015$ |

Table A11: Batch size ablation

| Batch size | Charville-Outcome | Skin-Imputation | SEA-Cell classification |
|---|---|---|---|
| 128 | $0.783 \pm 0.003$ | $0.791 \pm 0.007$ | $\mathbf{0.999 \pm 0.001}$ |
| 256 | $\mathbf{0.861 \pm 0.086}$ | $\mathbf{0.807 \pm 0.020}$ | $0.995 \pm 0.015$ |

