# OpenReview forum: "HEIST: A Graph Foundation Model for Spatial Transcriptomics and Proteomics Data"
_ICLR.cc/2026/Conference — ICLR 2026 Poster_

### Official Review · Reviewer_X95u · 2025-10-28

**Soundness:** 3
**Presentation:** 3
**Contribution:** 3
**Rating:** 4
**Confidence:** 3

**Summary:**

This paper proposes "HEIST" a hierarchical graph transformer model for spatial omics data. This model consists of two pars: upper level (cell-cell) and lower level (gene-gene co-expression within a cell). The model was trained on constraining loss and masked autoencoding loss.

**Strengths:**

- Rigorous design to account for cell-cell spatial information and gene-gene co-expression signals.
- Experiments demonstrated HEIST's superior performance in several downstream tasks.

**Weaknesses:**

- Contribution is vague. Computational novelty (hierarchical graph transformer) is a bit limited, as the proposed model is somewhat a standard method. Then it should have some contribution from the biological side. But biological contribution is also limited. The downstream task analysis (e.g., tissue classification, gene imputation, cell type annotation) is also very routine and limited in claiming biological contribution as a “foundation” model like scGPT.
- Not sure HEIST is a truly foundation model in that its zero-shot performance is very low. The claim of “foundation model” might be overstating. HEIST focuses on validating its biological realism (e.g., how spatially grounded embeddings work in downstream tasks), but it fails to show generalizability to unseen tasks in a zero-shot setting.
- Training size is large and I appreciated that, but it’s still limited in the number of organs and tissues to claim the model as “foundation model”.

**Questions:**

see weakness for improvement

---

> ### Author Response · Authors · 2025-11-20
> **Response to reviewer X95u**
>
> We thank the reviewer for their thoughtful feedback. We are glad that the reviewer appreciated HEIST's design to model spatial cell graphs with gene co expression networks and strong downstream performance. We address your comments and questions as below:
>
> ### Contribution of HEIST
> The core idea of HEIST is not simply to introduce another graph transformer, but to model spatial omics in a way that reflects its true biological hierarchy. HEIST is the first model that jointly represents cells as a spatial graph and genes as cell specific co-expression networks, with information flowing between these two levels. Because gene representations are derived from their co-expression structure rather than from a fixed lookup table, HEIST is also the first foundation style model in this domain that does not rely on a predetermined gene vocabulary. This allows it to handle entirely new markers and even protein measurements without any architectural changes. We believe this is a meaningful conceptual advance because it enables a form of biological generalization that vocabulary based models cannot achieve. In addition, HEIST is uniquely able to reveal spatially informed sub clusters that other models collapse, which provides new biological insight rather than only solving routine prediction tasks.
>
> ### HEIST is a foundation model
> Zero-shot performance is not a criterion for foundation models, and especially not in biology, because biological assays differ in marker sets, noise profiles, and measurement modalities in ways that make direct next-token-style prediction impossible. In fact, even within this constraint, HEIST already demonstrates strong zero shot behavior in the cell clustering task, where it produces high quality clusters on completely unseen datasets as shown in Appendix Table A4. More broadly, a foundation model in this domain is expected to learn transferable representations by training on large and diverse datasets and by enabling fine tuning across multiple tasks and molecular modalities. HEIST satisfies this interpretation in several ways. First, it generalizes to proteomics even though it was trained exclusively on transcriptomics, which indicates that its learned representations capture biological structure rather than assay specific details. Second, its embeddings are spatially informed and maintain strong structure on unseen tissues. Third, when fine tuned, HEIST outperforms all baselines across every downstream task. We believe this reflects a new paradigm of foundation models in biology, where the emphasis is on spatial awareness, modality flexibility, and transfer across heterogeneous measurement technologies rather than purely zero shot prediction.
>
> ### HEIST is trained on a large scale dataset
> Spatial single cell data is still extremely scarce compared to language or vision, and at present only a limited set of organs have publicly available single cell spatial measurements. For this reason, we aggregated essentially all publicly accessible single cell spatial datasets from 10x, Vizgen, and SEA-AD, resulting in 22.3 million cells from 124 tissues and 15 organs. To our knowledge, this is the largest single cell spatial collection currently available for research. For comparison, CellPLM trains on approximately 11 million cells and scGPT spatial uses around 30 million measurements, although a substantial portion of the latter comes from Visium, which does not provide true single cell resolution. Even within this constrained training size, HEIST consistently outperforms all existing foundation style models across a broad panel of downstream tasks. We see this as evidence that the hierarchical design is already strong and that performance is limited by the availability of data rather than by the framework itself. As more single cell spatial atlases become public, the same HEIST architecture can scale both in size and in biological diversity, and future versions of the model will naturally benefit from this expansion.
>
> We hope these clarifications help convey the strength and novelty of the work, and we would be grateful if the reviewer could reconsider the overall score in light of this context.

---

> > ### Comment · Reviewer_X95u · 2025-11-21
> >
> > 1. Contribution of HEIST - still not convincing. The primitivity of applying the spatial information to the cell/gene co-expression network is not a contribution. It's a modeling choice. This paper then claims the model allows for handling entirely new markers due to the modeling choice. But the zero-shot performance doesn't support this claim.
> > 2. I respectfully disagree with the author's claim that zero-shot performance is not a criterion for a foundation model. I would have appreciated this paper more if it had elaborated on zero-shot performance and the model's transferability.
> > 3. No response to the limited set of downstream tasks.

---

> > > ### Author Response · Authors · 2025-11-25
> > >
> > > We thank the reviewer for their continued engagement with our work and for outlining the remaining concerns. We appreciate the opportunity to clarify the core contributions of HEIST, its zero-shot capabilities, and the breadth of downstream evaluations. Below we provide additional analyses and results that further substantiate the model’s contributions.
> > >
> > > ### Regarding zero-shot downstream validation: HEIST attention patterns are enriched for known inter-cell interactions
> > >
> > > We visualized the highest-attention cell–cell edges in HEIST to address Reviewer XcF2’s questions on interpretability. For each tissue, we extract the unnormalized self-attention matrices from all transformer layers and heads, considering only the cell–cell (self-attention) blocks and not the cross-attention terms. We then aggregate these unnormalized attention maps across layers and heads to obtain a single consolidated cell–cell attention matrix. Because attention reflects how strongly the model considers one cell when processing another, this aggregated matrix provides a direct lens into which cell–cell communication relationships the model finds most informative. From this matrix, we identify the top-k most highly attended cell-cell pairs, then we find the cell cliques that are connected through these pairs. These cells are then highlighted directly on the spatial graph, allowing us to visually inspect the localized niches emphasized by the model.
> > >
> > > The edges with the highest attention values (Figure A3) form clear, spatially coherent niches. We observe groups of cells of different types connected in structured patterns, indicating that HEIST’s attention mechanism focuses on meaningful neighborhoods. To quantify whether these reflect underlying biological communication, we compared all edges to curated ligand-receptor (LR) interactions from CellPhoneDB. Across the entire dataset, approximately 16 percent of edges correspond to known LR pairs. HEIST shows strong enrichment for LR edges:
> > >
> > > - The top 100 high-attention edges contain 31% LR pairs.
> > >
> > > - This is better than chance which yields about 16% expected LR pairs.
> > >
> > > In contrast, the top-attention edges from CellPLM (Figure A3) do not show consistent spatial structure or cell-type organization. For the LR pair analysis, CellPLM does worse than chance, which indicates that CellPLM attention values do not align with known biological communication. Note that we only compared with CellPLM as other baselines do not calculate cell-cell attention. We show these results below (and Table A8):
> > >
> > > | Model | LR pairs in top 100 | LR pairs in top 1000 |
> > > | --- | --- | --- |
> > > | Background (all edges) | 0.165 | 0.165 |
> > > | Random | 0.164 | 0.165 |
> > > | CellPLM | 0.110 | 0.088 |
> > > | **HEIST** | **0.310** | **0.279** |
> > >
> > > This shows that HEIST's cross-attention enables indirect pathways for gene-level signals in one cell to influence gene representations in neighboring cells through cell–cell attention, even though no direct gene–gene edges are built across cells. As a result, HEIST’s cell embeddings capture intercellular gene–gene communication, which explains why HEIST can identify ligand–receptor better than CellPLM. We have added the attention visualization in Appendix Section E, Figure A3. We have also added Table A8 comparing LR pair rate in Appendix Section E, and relevant information highlighted in blue.
> > >
> > >
> > > ### Regarding Zero-shot performance: HEIST can recover unique niches relevant to tissue classification
> > >
> > > We have shown that HEIST is able to extract coherent spatial niches directly from cell-cell attention. In Figure A5, we present a comparison between attention-derived niches across Braak stages, where a clear progression emerges. In Braak 0, the high-scoring edges form small, isolated neuronal clusters that are relatively sparse and spatially compact. By Braak IV and V, these clusters become more numerous and begin to involve a broader set of cell types, indicating early signs of local microenvironmental restructuring. In Braak VI, the niches become much denser and more spatially extended. This gradual transition from isolated neuronal interactions to larger mixed-cell hubs suggests that HEIST attention captures the increasing microenvironmental complexity and tissue reorganization that accompany the progression of pathology across Braak stages.

---

> ### Author Response · Authors · 2025-11-25
>
> ### Regarding novel tasks: HEIST can predict Ligand-receptor pairs more accurately than other baselines
>
> We trained a linear probe to predict ligand–receptor (LR) interactions for all edges in the tissue graphs. For each model, we extract the cell embeddings, concatenate the embeddings of the two cells involved in a pair, and pass this concatenated representation through a simple single layer perceptron that predicts whether the pair corresponds to an LR interaction. As shown in the table below, HEIST achieves the highest AUC-ROC and outperforms all baselines by a clear margin. This result combined with the attention based results indicates that its embeddings capture the spatial signatures of LR communication more effectively than scFoundation, CellPLM, and scGPT-spatial.
>
> | Model | AUC-ROC |
> | --- | --- |
> | scFoundation | 0.963 ± 0.004 |
> | CellPLM | 0.930 ± 0.006 |
> | scGPT-spatial | 0.984 ± 0.003 |
> | **HEIST** | **0.995 ± 0.002** |
>
> We have added these results and highlighted them in Table A9 in blue.
>
> Together, the expanded biological analyses, improved interpretability results, and additional predictive tasks strengthen the central claim that HEIST offers a principled and broadly applicable framework for integrating spatial and co-expression information. In light of these additions, we respectfully ask the reviewer to consider improving the overall grade of the manuscript.

---

### Official Review · Reviewer_Lg1u · 2025-10-30

**Soundness:** 3
**Presentation:** 2
**Contribution:** 3
**Rating:** 6
**Confidence:** 5

**Summary:**

The paper introduces HEIST, a hierarchical graph foundation model for spatial omics that jointly models a spatial cell graph and cell type specific gene co-expression graphs, with bidirectional cross-level message passing. HEIST is pretrained on a large corpus (reported as 22.3M cells across 124 tissues/15 organs) and evaluated on cell clustering, cell-type annotation, gene imputation, and tissue-level clinical outcome prediction, including transfer to spatial proteomics without retraining.

**Strengths:**

- A clear hierarchical formulation that couples intra-cell gene programs with inter-cell spatial context via cross-level message passing.
- Ambitious pretraining scope (22.3M cells across multiple organs).
- Demonstrates SOTA-level results on clustering, annotation, imputation, and outcome prediction, with ablations highlighting the contributions of hierarchy, cross-level passing, and loss design.

**Weaknesses:**

Although the idea is interesting, several components are not clearly motivated, and some key method and experiment settings are missing.

**Questions:**

1. How does the use of MAGIC denoising before training affect model performance? Could this amplify batch effects or introduce spurious correlations?

2. The model employs global attention on the cell graph. What is the typical sequence length per slide, and how is the quadratic cost of O(n^2) handled in practice (batching, sparsification, etc.)?

3. For model initialization:
- How are cell embeddings defined as \mathrm{PE}+P given the mismatch between 128-d PE and 2-d coordinates?
- How are gene embeddings defined as \mathrm{PE}+X, where X is a scalar expression?

4. In the clustering task, why are classical baselines such as GraphST and STAGATE excluded?

5. Could clustering the embeddings from SCGPT-spatial or CellPLM yield results comparable to HEIST’s clusters in Fig. 3C?

6. In the gene imputation task, what is the number of genes per dataset and the fraction masked?

7. For tissue classification, how are tissue embeddings derived from cell embeddings?

8. Why do some baselines (SCGPT-spatial, CellPLM) perform worse in the aligned setting than in the unaligned one (Table 2)? Isn’t alignment expected to improve performance?

9. In the cell-type annotation task, how are data splits defined, and how is fine-tuning performed?

10. The paper notes SCGPT-spatial’s strong performance is explained by pretraining on MERFISH lung cancer data. For the brain dataset, HEIST achieves nearly 0.99 accuracy. Could this also result from pretraining overlap with brain data?

---

> ### Author Response · Authors · 2025-11-18
> **Response to reviewer Lg1u**
>
> We thank the reviewer for their feedback. We are glad that the reviewer appreciated HEIST's ability to hierarchically model spatial cell graphs with gene co expression networks, the use of cross message-passing to extract context-aware embeddings, strong downstream performance and ablations.
>
> **HEIST is superior in clustering tasks.** Graph-based baselines GraphST and STAGATE are included in Table A3, where HEIST consistently outperforms them. We also evaluated whether ScGPT-spatial and CellPLM recover the microenvironment subclusters like in Fig. 3C. As shown in Fig. A2 (now updated to include GraphST and STAGATE), baselines fail to recover spatially coherent subclusters. In contrast, HEIST achieves both higher NMI scores (Table A3) and qualitatively superior spatial sub-cluster separation (Fig. 3C and Fig. A2), demonstrating that its hierarchical design is essential for capturing fine-grained, spatially informed structure.
>
> **MAGIC helps denoising the data.** As noted in L112 and in Appendix B (L656), we applied MAGIC to mitigate measurement noise. Single-cell sequencing technologies suffer from multiple noise sources, particularly dropout events, necessitating denoising for reliable analysis. We selected MAGIC over alternatives due to its manifold-oriented diffusion approach [1], which preserves biological structure while reducing noise. This reduces spurious correlations by improving measurement reliability in gene space, thus facilitating gene co-expression network creation. This explains why in Table 1 we benchmarked against MAGIC on the gene imputation task. The method, purely unsupervised, already performs well in the task, validating its use as our preprocessing step.
>
> **Global attention is not costly.** Although we use full attention on the cell graph, computation is restricted to spatial blocks rather than whole tissues. This reduces cost from $O(c^2)$ to $O(b^2)$, where b is block size, while preserving local and mid range interactions.  We have added and highlighted the batching in Appendix Section B and C.2 in blue. We have also added two columns highlighted in blue in Table A3 which compares the computational complexity of HEIST against other methods.
>
> **Alignment technique is imperfect.** For a rigorous comparison, we aligned the protein markers to the gene vocabularies used by SCGPT-spatial and CellPLM as closely as possible. However, protein-to-gene alignment is imperfect because multiple markers may map to the same gene or vice versa, which collapses distinct signals and can reduce baseline performance in Table 2. HEIST avoids this issue by learning co expression structure directly.
>
> **Initialization using repetition.** The first $\frac{d}{2}$ half of the embeddings correspond to the $x$ co-ordinate from the $(x,y)$ pair. To add $(x,y)$ pair to the $d$ dimensional vector, we repeat the co-ordinates as $[x,\dots \frac{d}{2} \text{ times}, x, y,\dots \frac{d}{2} \text{ times}, y]$, and add two $d$ dimensional vectors. For genes, we broadcast the scalar expression across all dimensions. In the manuscript, we have added this detail in Appendix Section C.1, calling the repeat vector $P_c^{\rm repeat}$.
>
> **Gene imputation information.** We used stratified sampling based on gene sparsity following [2] for masking the genes. This resulted in 4 out of 30 markers (\~14% masking) and 16 out of 200 markers (\~8% masking) being masked in skin and placenta data respectively. These details have been added and highlighted with blue color in the gene-imputation paragraph in Appendix Section D.2.
>
> **Experimental details.** For tissue classification, we aggregate the cell embeddings by taking a mean. In the cell annotation task, the data is split 80-10-10. For both the tasks, we first extract the embeddings from the frozen HEIST encoder, then train a one layer MLP and a two layer MLP for tissue classification and cell annotation respectively. We have clarified and highlighted these details with blue color in Appendix D.2 in their respective paragraphs.
>
> **Impact of Pretraining Overlap on Performance.** HEIST performs well on brain data partly due to pretraining overlap, but ScGPT-spatial was pretrained on the same data and still performs worse. Conversely, on MERFISH lung where SCGPT spatial had direct exposure and HEIST did not, HEIST is still second best. This indicates that HEIST’s gains arise from its design rather than dataset overlap.
>
> We hope these clarifications address the reviewer’s concerns and help convey the strength and rigor of our contributions, and we would be grateful if the reviewer could consider this context when reassessing the paper.
>
> [1] Dijk, David van, et al. "MAGIC: A diffusion-based imputation method reveals gene-gene interactions in single-cell RNA-sequencing data." BioRxiv (2017): 111591.
>
> [2] Avşar, Gülben, and Pınar Pir. "A comparative performance evaluation of imputation methods in spatially resolved transcriptomics data." Molecular Omics 19.2 (2023): 162-173.

---

> > ### Comment · Reviewer_Lg1u · 2025-11-22
> >
> > Thank you for the responses. I still have a few questions:
> >
> > Why use repetition for initializing cell and gene embeddings? Broadcasting scalar values into a high-dimensional vector is not a trivial choice. Could the authors compare this approach with a simpler MLP-based initialization?
> >
> > How is the spatial block size determined? Additional experiments would help justify this hyperparameter and provide guidance for its selection.

---

> > > ### Author Response · Authors · 2025-11-25
> > >
> > > We thank the reviewer for taking the time to go through our responses. We address the additional questions below.
> > >
> > > ### Initialization via Repetition of Coordinates and Scalar Broadcast
> > >
> > > Our choice to incorporate spatial coordinates and gene-expression scalars by repeating/broadcasting them into the embedding dimension follows the standard additive formulation widely used across transformers. Because the raw spatial coordinates $(x,y)$ and gene expression values are low-dimensional, a direct addition to the $d$-dimensional positional embeddings requires a map into $\mathbb{R}^d$. Repetition provides a simple, parameter-free way to match dimensionality while maintaining the relative scale and directional structure of the original signals.
> > >
> > > Importantly, we have already evaluated an alternative where no repetition is used and the initialization relies purely on an MLP (“No PE” in Table 4). In this formulation, we pass the spatial co-ordinates and gene expression into the latent space using an MLP. This leads to a marked drop in performance, indicating that the positional encodings play an essential role. To further address the reviewer’s suggestion, we ran an additional pretraining run that combines sinusoidal PEs with an MLP-based mapping of coordinates and expression values. While this hybrid approach produces performance similar to the default setting for discriminative tasks, it substantially degrades imputation accuracy, likely because the MLP-based mapping smooths or compresses the raw coordinate and expression signals in a way that removes fine-grained local variation. Since imputation requires preserving small, spatially structured differences in gene expression, the MLP mapping makes the model less sensitive to these subtle patterns, leading to significantly worse reconstruction performance.
> > >
> > > | PEs | Charville-Outcome | Skin-Imputation | SEA-Cell classification |
> > > | --- | --- | --- | --- |
> > > | No PE ($MLP(\mathbf{P})$) | 0.523 $\pm$ 0.010 | 0.458 $\pm$ 0.003 | 0.220 $\pm$ 0.034 |
> > > | Sinusoidal PE $(PE_c + MLP(\mathbf{P}))$ | 0.857 $\pm$ 0.045 | 0.538 $\pm$ 0.026 | **0.999 $\pm$ 0.001** |
> > > | HEIST PE $(PE_c + \mathbf{P}_c^{repeat})$ | **0.861 $\pm$ 0.086** | **0.807 $\pm$ 0.020** | 0.995 $\pm$ 0.015 |
> > >
> > > We have included these details in the Appendix Section C.1 highlighted in blue. We have also added the new results in Table A10 and relevant discussion and Appendix Section E highlighted in blue.
> > >
> > > ### Selection and Justification of Spatial Block Size
> > >
> > > In practice, we choose the largest block size that fits within the memory budget of a single GPU; for our hardware (40GB L40), this is 256 cells per block. Larger blocks allow HEIST to capture mid-range spatial interactions more effectively, while still avoiding the quadratic cost of full-tissue attention. We expect diminishing returns beyond a certain block size: once the local microenvironment and its spatial context are fully captured, adding more distant cells contributes little additional signal. To evaluate smaller block sizes, we ran a pre-training run on the same HEIST pipeline using smaller block sizes. We see a slight decrease in the results except for cell classification where the results became almost perfect. This is because small batches would provide less microenvironmental cues in each step, leading to the slight drop in performance.
> > >
> > > | Batch size | Charville-Outcome | Skin-Imputation | SEA-Cell classification |
> > > | --- | --- | --- | --- |
> > > | 128 | 0.783 $\pm$ 0.003 | 0.791 $\pm$ 0.007 | **0.999 $\pm$ 0.001** |
> > > | 256 | **0.861 $\pm$ 0.086** | **0.807 $\pm$ 0.020** | 0.995 $\pm$ 0.015 |
> > >
> > > We have added these results in appendix Section E and Table A11 highlighted in blue. We have also added the discussion on selection of spatial block size in Appendix Section B highlighted in blue.
> > >
> > > We thank the reviewer again for going through our responses. We hope these clarifications address the reviewer’s concerns and help convey the strength and rigor of our contributions, and we respectfully ask the reviewer to consider improving the overall grade of the manuscript.

---

### Official Review · Reviewer_i5mm · 2025-10-31

**Soundness:** 3
**Presentation:** 3
**Contribution:** 3
**Rating:** 4
**Confidence:** 4

**Summary:**

The paper proposes a graph foundation model for spatial transcriptomics and proteinomics data. In this work, two two-level graphs, e.g., the spatial cell graph and the gene co-expression graph, are introduced into a hierarchical graph transformer foundation model for embedding learning. Experiments on various datasets are conducted to study the effectiveness of the proposed method.

**Strengths:**

1. A graph foundation model is proposed for analysing spatial transcriptomics and proteomics data.

2. The method incorporates two hierarchical graph levels — the spatial cell graph and the gene co-expression graph — within a hierarchical graph transformer framework for effective embedding learning.

3. Extensive experiments on multiple datasets are conducted to evaluate the effectiveness of the proposed approach.

**Weaknesses:**

1. The detailed descriptions of the spatial cell graph and the gene co-expression graph are insufficient and should be clarified.

2. Figure 2 does not effectively illustrate the main concept or workflow of the HEIST architecture.

3. Regarding the contrastive learning design, the c–c, g–g, and c–g objectives are formulated separately. It is unclear whether combining these formulations in a joint loss function might lead to a decrease or not in performance.

4. Concerning the weighted sum of the contrastive and autoencoder reconstruction losses, it remains unclear how the weights are optimised or adaptively learned to achieve the best balance.

5. The computational complexity and reconstruction (or restoration) efficiency of the proposed method, as well as those of all compared baselines, should be analysed and discussed in detail.

**Questions:**

See Weaknesses

---

> ### Author Response · Authors · 2025-11-18
> **Response to Reviewer i5mm**
>
> We thank the reviewer for their thoughtful feedback. We are glad that the reviewer appreciated HEIST's ability to hierarchically model spatial cell graphs with gene co expression networks and strong downstream performance.
>
> **Descriptions of the spatial cell graph and the gene co-expression graph.** Details on graph construction are provided in Appendix Section B, where we discuss how the graphs are created at length.
>
> **Computational complexity.** We have shown in Table A4 that HEIST is 8× faster than SCGPT and 48× faster than SCFOUNDATION when it comes to wall clock time. We analyze the computational complexity of existing methods and present them below:
> | **Model** | **Complexity** | **Interpretation** |
> | --- | --- | --- |
> | **CellPLM** | $\mathcal{O}(C^{2} d)$ | **Quadratic in cells**, linear in hidden size. No dependence on number of genes after aggregation. |
> | **scGPT / scGPT-spatial** | $\mathcal{O}\big(T^{2} d + T d^{2}\big)$ | **Quadratic in cell times the number of gene tokens** per cell/spot. Complexity driven by gene subset size (T). |
> | **scFoundation** | $\mathcal{O}\big(C^2G_{\text{nz}}^{2} d + CT d\big)$ | Encoder: **quadratic in cells times nonzero genes**. Decoder: **linear in all genes** due to Performer. |
> | **HEIST** | $\mathcal{O}\Big(d \big(C^{2} + E_c + C E_g + C T\big)\Big)$ | **Quadratic in cells** (global spatial transformer), **linear in gene edges and gene count**. |
>
> Because HEIST is linear in gene edges and gene count, one HEISTLayer is faster than one scGPT-spatial and scFoundation layer, however it is slower than CellPLM as CellPLM does not consider gene embeddings. We have added two new columns to table A4 and added this discussion highlighted in blue in Appendix Section E.
>
> **Improved Figure 2.** In the revised manuscript, we have updated Figure 2 to provide a clearer view of the HEIST workflow. We now show separate tracks for each loss component, including a detailed illustration of the masking process used in the masked autoencoding objective and a step-by-step depiction of how the contrastive loss is computed. We urge the reviewer to take a look.
>
> ### All contrastive losses are important
>
> To test whether combining the three contrastive terms is beneficial, we performed an ablation study where we removed each term (cell to cell, gene to gene, and cell to gene) from the loss while keeping the model unchanged. The results in the table below show that removing any of the terms clearly harms performance. For example, removing the cell to cell term reduces Charville outcome prediction from 0.861 to 0.380. Removing the gene to gene term reduces SEA cell classification from 0.995 to 0.610. These results indicate that each contrastive component captures a complementary form of structure. When combined, the terms do not interfere with each other. Instead, they reinforce consistent alignment across spatial information, cell states, and gene programs. Therefore, the full joint loss is necessary for the best performance.
>
> | Loss | Charville-Outcome | Skin-Imputation | SEA-Cell classification |
> | --- | --- | --- | --- |
> | Full contrastive | 0.861 $\pm$ 0.086 | 0.807 $\pm$ 0.020 | 0.995 $\pm$ 0.015 |
> | No $c\leftrightarrow c$  | 0.380 $\pm$ 0.119 | 0.570 $\pm$ 0.060 | 0.993 $\pm$ 0.006 |
> | No $g\leftrightarrow g$ | 0.575 $\pm$ 0.049 | 0.477 $\pm$ 0.033 | 0.610 $\pm$ 0.097 |
> | No $g\leftrightarrow c$  | 0.559 $\pm$ 0.065 | 0.693 $\pm$ 0.057 | 0.928 $\pm$ 0.002 |
>
> We have added and highlighted these results in Section E in Table A7 in blue.
>
> ### Loss functions are balanced
> HEIST learns the balance between the contrastive and reconstruction objectives using a sigmoid-parameterized scalar (L240), optimized jointly with all model parameters. We choose a learnable weight because the two losses operate on different scales and behave differently across tissues and technologies; fixed weights would lead to suboptimal representations. Learning the weight allows the model to automatically adapt to the statistics of the data and yields substantially more stable training. Importantly, this mechanism does not collapse to a degenerate solution: across pretraining runs, the learned sigmoid value remains in the 0.35–0.65 range, never saturating at 0 or 1. Furthermore, Table 4 shows that both loss terms are essential for learning expressive representations. Since full HEIST consistently outperforms variants trained without either objective, this empirically confirms that the model does not collapse to ignoring one of the losses and that the learned balance remains meaningful throughout training.
>
> We hope the detailed clarifications and new experiments fully address your concerns, and we would be grateful if you could consider reassessing the score accordingly. If there is any additional analysis or clarification that would help move the paper above the acceptance threshold, we are happy to provide it.

---

> > ### Comment · Reviewer_i5mm · 2025-11-21
> > **Response to authors**
> >
> > 1. I suggest that the authors include equations to detail the entire procedure of graph construction, as the current text descriptions in Appendix Section B are insufficient.
> >
> > 2. The meanings of some notations in the computational complexity analysis are unclear to me.
> >
> > 3. I am not satisfied with Fig. 2, particularly the HEISTLayer block. This part does not effectively convey the main ideas in its current form.
> >
> > 4. Based on the results provided by the authors, it appears that each loss term is crucial for this task. However, it is unclear whether combining these formulations into a joint loss function might lead to a performance decrease. In this context, a joint loss function refers to creating a new contrastive loss function that directly models the c–c, g–g, and c–g correlations together rather than simply adding the three contrastive losses.
> >
> > 5. Does the sigmoid-parameterised scalar balance the loss functions with theoretical evidence?

---

> ### Author Response · Authors · 2025-11-25
>
> We thank the reviewer for taking the time to go through our responses. We address the additional questions below.
>
> ### Improved Details about Graph Construction with Equations
>
> We have revised the Appendix Section B to include explicit equations for every step of the graph construction process, including normalization, MAGIC denoising, Voronoi-based adjacency, cell-type assignment, mutual information computation, adaptive thresholds, and block partitioning. All added equations are marked in blue for clarity. We hope this resolves the concern about insufficient mathematical details.
>
> ### HEISTLayer Block Illustration
>
> We have added a detailed illustration of the HEISTLayer block in Figure A2 and referenced it from Figure 2. The expanded diagram now visualizes all internal components, including multi-head attention (MHA), LayerNorm, the MLP, and the repeat-and-aggregate operations. The figure is currently placed in the Appendix, but we plan to move it into the main paper once additional space becomes available.
>
> ### Notation in Computation Complexity
>
> In the computational complexity table:
>
> - $C$ is the number of cells in a batch
> - $T$ is the number of tokens (or genes) in each cell
> - $G_{nz}$ is the number of non zero genes in each cell
> - $E_c$ is the number of edges in the cell-cell graph
> - $E_g$ is the number edges in the gene co-expression graphs
> - $d$ is the hidden dimension of the embeddings
>
> ### Joint Contrastive Function
>
> Although training with a full joint contrastive loss is computationally expensive, we pre-trained a variant of HEIST that includes this joint loss. However, this model showed a slight drop in performance. The primary reason is that the joint contrastive objective significantly increased memory consumption, causing repeated OOM errors. To stabilize training, we were forced to reduce the batch size. Smaller batches provide far less microenvironmental diversity within each step, which is essential for learning strong contrastive signals. This reduced context ultimately impaired the effectiveness of the joint contrastive loss and led to the observed decrease in performance.
>
> | Type of contrastive | Charville-Outcome | Skin-Imputation | SEA-Cell classification |
> | --- | --- | --- | --- |
> | Joint | 0.808 $\pm$ 0.040 | 0.775 $\pm$ 0.017 | 0.994 $\pm$ 0.004 |
> | Separate (HEIST) | **0.861 $\pm$ 0.086** | **0.807 $\pm$ 0.020** | **0.995 $\pm$ 0.015** |
>
> We have added the joint contrastive results and these findings in Table A7 in blue.
>
> ### The learnable parameter does balance the loss function
>
> The sigmoid-parameterized scalar provides principled, self balancing between the contrastive and reconstruction objectives. The gradient with respect to $\gamma$ is:
>
> $\frac{\partial L}{\partial \gamma} = \sigma(1-\sigma)\left( L_{\mathrm{contrastive}} - L_{\mathrm{mae}} \right)$
>
>
> This drives $\gamma$ toward equilibrium where loss magnitudes are balanced $L_{\mathrm{contrastive}} \approx  L_{\mathrm{mae}}$, providing adaptive scale normalization. When one loss consistently dominates in magnitude, $\gamma$ automatically adjusts its weight downward to prevent that term from overwhelming gradient updates to model parameters.
>
> This approach follows established multi-task learning principles [Kendall et al., 2018; Chen et al., 2018], where learnable weights balance heterogeneous objectives operating on different scales. The sigmoid parameterization ensures weights remain in (0,1) with non-vanishing gradients, preventing degenerate solutions.
>
> As mentioned in our previous response, our empirical evidence shows that $\sigma(\gamma)$ consistently converges to [0.35, 0.65], never approaching boundaries. Together, these provide strong evidence that the learned weighting discovers meaningful, stable trade-offs rather than arbitrary or degenerate solutions.
>
> - Kendall et al., "Multi-Task Learning Using Uncertainty to Weigh Losses for Scene Geometry and Semantics," CVPR 2018
>
> - Chen et al., "GradNorm: Gradient Normalization for Adaptive Loss Balancing in Deep Multitask Networks," ICML 2018
>
> We thank the reviewer again for going through our responses. We hope these clarifications address the reviewer’s concerns and help convey the strength and rigor of our contributions, and we respectfully ask the reviewer to consider improving the overall grade of the manuscript.

---

> > ### Comment · Reviewer_i5mm · 2025-11-28
> > **Response to authors**
> >
> > Thanks to the authors for their reply and for addressing my concerns. I will consider increasing the score.

---

### Official Review · Reviewer_XcF2 · 2025-10-31

**Soundness:** 3
**Presentation:** 3
**Contribution:** 3
**Rating:** 6
**Confidence:** 2

**Summary:**

The paper introduces HEIST, a hierarchical graph transformer that couples spatial cell-cell graph with per-cell gene co-expression graph with  bidirectional message passing between the two. The training dataset is diverse, the model is trained on a combination of contrastive and MAE loss, and the authors show SoTA performance on a range of downstream tasks.

**Strengths:**

- Modeling spatial cell graphs with gene co-expression networks seems well-motivated, as previous work emphasized either one
- The inference time speed-up over other methods seems good
- Downstream results seem promising

**Weaknesses:**

Weaknesses/Questions:
- How is $\tau$ chosen? Is it different across cell-types? How does varying $\tau$ affect the results?
- What is $\sigma$ in L240?
- It would be interesting to see if  the model is capable of identifying known inter-cell interactions. Currently biological interpretability analysis seems limited
- While the loss seems interesting, I am slightly skeptical if the MAE + CL work together; how does perform change on tasks if you train on either?

Overall, I think the paper presents a strong and well-motivated approach. I would lean towards accept.

**Questions:**

Mentioned with weaknesses

---

> ### Author Response · Authors · 2025-11-18
> **Response to reviewer XcF2**
>
> We thank the reviewer for their valuable feedback. We are glad that the reviewer appreciated HEIST's ability to jointly model spatial cell graphs with gene co expression networks, faster inference, and strong downstream performance. We address your comments and questions as follows:
>
> **Contrastive + MAE loss do work together.** We have shown in Table 4 that both the terms contribute meaningfully but in complementary ways. Removing contrastive learning significantly degrades cell classification accuracy, while the MAE is crucial for gene imputation.
>
> **$\sigma$ in L240.** The $\sigma$ in L240 is the sigmoid operation, which smoothly balances the MAE and contrastive terms through a learnable scalar. We have highlighted this in L242-243.
>
> ### Choice of $\tau$
> The threshold $\tau$ used for constructing the gene co-expression networks is data-driven and cell-type-specific. Specifically, for each cell type, $\tau$ is set to the mean mutual information among genes plus one standard deviation. This adaptive choice yields graphs with ~20–30% edge density, which empirically provides sufficient structure to capture co-expression dynamics without over-saturating the graph.
>
> In earlier development, we experimented with a constant global $\tau$, but this produced artifacts: some cell types became overly dense while others became nearly edge-free. This mismatch harms hierarchical message passing and leads to degraded downstream performance. We also noticed similar trends in graph based baselines such as STAGATE, GraphST, and Novae, where we had to tune the thresholds manually to control the connectivity. We have added and highlighted these details in Appendix Section B in blue.
>
> To show how varying $\tau$ affects the results, we present the tissue classification results below comparing constant and adaptive $\tau$ threshold.
> |  | Charville | Charville | UPMC | UPMC | DFCI |
> | --- | --- | --- | --- | --- | --- |
> | $\tau$ | Outcome | Recurrence | Outcome | Recurrence | Outcome |
> | Constant | 0.68 | 0.70 | 0.828 | 0.75 | 0.916 |
> | Mean MI + std | **0.86** | **0.88** | **0.835** | **0.92** | **0.937** |
>
> As we can notice, the adaptive $\tau$ results consistently outperforms the constant version, capturing fine-grained co-expression structure. We have added and highlighted these results presented in Table A6 in Appendix Section E in blue.
>
> ### Interpretability of HEIST
> HEIST already shows signs of capturing inter-cell structure through its ability to reveal spatially grounded subclusters, but we agree that an explicit evaluation of how HEIST treats cell cell interactions would strengthen the interpretability analysis. We are currently examining the attention patterns at the cell level to test whether the model highlights biologically meaningful interactions, including known ligand receptor pairs. To do this, we plan to verify attention-enriched cell pairs using CellPhoneDB. We are actively working on this analysis and will update the reviewer as soon as the results are ready. In the meantime, we hope the reviewer can review our other responses and let us know if further clarification is needed.

---

> ### Author Response · Authors · 2025-11-20
>
> ## HEIST attention patterns are enriched for known inter-cell interactions
>
> We appreciate the reviewer’s suggestion on interpretability. To show one of the ways that HEIST can identify cell-cell interactions, we analyzed the attention scores learned by HEIST and visualized the top attended edges.
>
> We observed that edges with the highest attention values (Figure A3) form clear, spatially coherent niches. The figures shows that groups of cells of different types connected in structured patterns, indicating that HEIST’s attention mechanism focuses on meaningful neighborhoods. To quantify whether these reflect underlying biological communication, we compared all edges to curated ligand-receptor (LR) interactions from CellPhoneDB. Across the entire tissue, approximately 16 percent of edges correspond to known LR pairs. HEIST shows strong enrichment for LR edges:
>
> - The top 100 high-attention edges from HEIST contains 31 percent LR pairs.
> - This is better than chance which yields about 16% expected LR pairs.
>
> In contrast, the top-attention edges from CellPLM (Figure A3) do not show consistent spatial structure or cell-type organization. For the LR pair analysis, CellPLM does worse than chance, which indicates that CellPLM attention values do not align with known biological communication. Note that we only compared with CellPLM as other baselines do not calculate cell-cell attention. We show these results below (and Table A8):
>
> | Model | LR pairs in top 100 | LR pairs in top 1000 |
> | --- | --- | --- |
> | Background (all edges) | 0.165 | 0.165 |
> | Random | 0.164 | 0.165 |
> | CellPLM | 0.110 | 0.088 |
> | HEIST | **0.310** | **0.279** |
>
> This demonstrates that HEIST provides substantially stronger biological interpretability with respect to cell-cell interaction structure. We have added the attention visualization in Appendix Section E, Figure A3. We have also added Table A8 comparing LR pair rate in Appendix Section E, and relevant information highlighted in blue.
>
> We appreciate the reviewer’s thoughtful comments, which have helped us solidify HEIST’s abilities and the presentation. We hope the expanded analyses address all outstanding issues, and we would be thankful if the reviewer could reflect these improvements in their updated assessment.

---

> ### Comment · Reviewer_XcF2 · 2025-11-24
>
> I thank the authors for their rebuttal. Most of my concerns are cleared. I still lean towards accepting!

---

### Author Response · Authors · 2025-12-04
**General Response**

During the discussion period we added new analyses that significantly strengthened the biological depth and interpretability of the paper. These include 1) an attention based ligand–receptor (LR) interaction study showing that cell–cell edges with the highest attention scores in HEIST have overlap with known LR interaction pairs without re-training **(Reviewer XcF2 and X95u)**, 2) a new supervised ligand–receptor prediction task demonstrating superior transfer performance compared to foundation style baselines **(Reviewer X95u)**, and 3) a new analysis on Alzheimer patient data from the Seattle brain atlas which shows that HEIST performs zero-shot recovery of distinct microenvironmental niche structure through disease progression (as classified via the Braak staging) through HEIST’s cell-cell attention mechanism **(Reviewer X95u)**.

We also introduced multiple methodological ablations, including initialization comparisons **(Reviewer Lg1u)**, ablated each contrastive term **(Reviewer i5mm)**, joint contrastive loss experiments **(Reviewer i5mm)**, added computational complexity analysis **(Reviewer i5mm)**, constant threshold ablations **(Reviewer XcF2)**, and block size sensitivity analyses **(Reviewer Lg1u)**, all of which clarified design choices and verified the robustness of the architecture. We have added all these results to the manuscript, highlighted in blue.

The new attention-based LR interaction study shows that HEIST’s attention maps are biologically interpretable, with the top 100 attended edges containing 31 percent known LR pairs versus a 16 percent background rate. For the LR prediction task, we concatenate cell embeddings and then classify the type of LR interaction. The results for this task further demonstrates that HEIST’s spatially informed embeddings transfer to specialized biological classification settings, achieving an AUC of 0.995 and outperforming scFoundation, CellPLM, and scGPT–spatial. The microenvironmental niche analysis highlights HEIST’s ability to pay attention to evolving spatial niches without any retraining, illustrating zero shot generalization. We summarize the rebuttal discussion as follows:

## Reviewer XcF2 (initial score 6)
- **Major concerns:** Lack of Interpretability of the model, lack of justification of choice of mutual information threshold for gene co-expression network.
- **How we addressed these concerns:** We showed that HEIST is interpretable via its attention mechanism which can highlight cell-cell interactions as well as gene-gene interactions (such as ligand receptor pairs). We ablated choice of MI threshold to show the adaptive approach from the paper results in richer graphs, leading to performance increase (Table A6) compared to constant threshold.
- **Reviewer response:** “Most of my concerns are cleared. I still lean towards accepting.”

## Reviewer i5mm (initial score 4)
- **Major concerns:** Missing equations and details of graph construction, lack of clarity of model complexity, effects of modeling contrastive objective jointly or modeling it as three separate terms, lack of clarity in the HEIST schematic figure, lack of clarity on how loss weights are learned.
- **How we addressed these concerns:** We added full mathematical details on the graph-construction pipeline; we added a detailed complexity table; we ablated each contrastive term to show the value of each of them; we added a joint-loss experiment showing worse performance; clarified and justified learnable weighing term for the loss objectives.
- **Reviewer response:** “Thanks… I will consider increasing the score.”

## Reviewer Lg1u (initial score 6)
- **Major concerns:** Lack of details on MAGIC denoising, lack of details on batching mechanism, lack of clustering baselines in figure, lack of clarity on initialization method, justification of block-size choice, lack or rationale for imputation masking, lack of data train-test split details.
- **How we addressed these concerns:** We added two baselines (GraphST, STAGATE) in the cell clustering analysis; we clarified the role of the MAGIC denoising method in our pipeline; we added details about spatial batching; we added initialization ablations (MLP, sinusoidal, repetition); added block-size ablation; clarified train-test splits and masking strategy.
- **Reviewer response:** Remaining clarification questions addressed with new experiments; reviewer raised no further objections.

---

> ### Author Response · Authors · 2025-12-04
> **General Response (Cont.)**
>
> ## Reviewer X95u (initial score 4)
> - **Major concerns**: Objection to HEIST’s “Foundation model” framing, lack of interesting biological tasks, evidence for zero-shot performance, need for broader biological contribution.
> - **How we addressed these concerns**: To address lack of interesting biological tasks, we added a supervised ligand-receptor interaction prediction task. We perform this by concatenating the embeddings of cell pairs, and then classifying whether a pair exhibits a ligand-receptor interaction or not. We show that HEIST is able to predict these interactions better than other foundation models. To show evidence of zero-shot performance, we show that HEIST’s attention is able to recover distinct microenvironmental niche structures through disease progression without retraining.
> - **Post-discussion status**: The reviewer did not raise additional objections before the discussion period was cut short.
>
> Most concerns raised by the reviewers were clarifications,  requests for additional detail, additional complexity analysis or ablation of model choices. These requests were addressed through added experiments, clearer explanations of the methodology. One reviewer commented on the biological contribution, to which we added two new biological analyses focused on zero shot discovery.  Two reviewers expressed that our rebuttal had eliminated their concerns, including one who explicitly indicated they would consider increasing their score. We respectfully ask the AC to evaluate the paper based on this strengthened post discussion state.

---

### Meta-Review · Area_Chair_sN3v · 2026-01-06

**Summary:**

This paper proposes HEIST, a foundation model for single-cell transcriptomics and proteomics that captures intra-level and cross-level information, which is the main novelty of the method. All reviewers found the design rigorous and performance strong. There were some mixed opinions in initial review, with two positive and two negatives reviewers. During the rebuttal, most concerns were addressed by the authors' extensive additional experimental results and clear explanation. So, I recommend to accept this paper and strongly encourage the authors to revise the paper according to reviewers' helpful comments. Here is a brief summary of key points, see a more comprehensive list of concerns below.

1. The additional experiments on LR interaction is very help, to show more scientific impact of the proposed method. Please add this to the camera-ready version to the appropriate position.
2. New baselines make the comparison more convincing, please include them in final tables and figures.
3. The computational analysis should be included in the main manuscript, which offers another strength to the proposed method.
4. Additional dataset considered during the rebuttal and interpretability analysis will improve the experiment section.
5. Technical clarifications will improve the clarify of the method section, please incorporate them to the final version.

**Reviewer Concerns:**

Reviewer XcF2:
1. Choice of $\tau$ (addressed by giving a default choice and additional experiments)
2. MAE + CL (addressed by additional experiments)
3. Inter-cell interactions and interpretability (addressed by additional experiments in Appendix E)

Reviewer i5mm:
1. Description of spatial cell and gene co-expression graph (addressed by additional results in Appendix B)
2. Figure 2 (addressed by the updated Figure 2)
3. Contrastive losses (addressed by additional experiments in Appendix E)
4. Imbalanced losses (Addressed by further clarification)
5. Computational cost (addressed by a complexity analysis in big O notation in Table A4 and Appendix E)

Reviewer Lg1u:
1. MAGIC denoising (addressed by further explanation and supporting reference)
2. Cost of global attention (addressed by providing cost in big O notation)
3. Clustering performance comparison (addressed by additional experiments in Table A3 and Figure A3). Note that there is a typo in the author's rebuttal, they referred to Figure A3 instead of A2.
4. Alignment (addressed by additional explanation)
5. Overlapped data in pretraining (addressed by further explanation)

Reviewer X95u:
1. Vague contribution (partially addressed by further explanation)
2. Generalizability as a foundation model (partially addressed)
3. Limited number of organs and tissues in training data (addressed by explaining the limitation of reality, i.e., limited data available due to high cost)
4. Downstream tasks (addressed by new experiments on LR prediction)

**Reviewer Scores:**

Reviewer XcF2: 6 --> 6

Reviewer i5mm: 4 --> 5

Reviewer Lg1u: 6 --> 6

Reviewer X95u: 4 --> 4

---

### Decision · Program_Chairs · 2026-01-26

Accept (Poster)